# Chemically-defined and scalable culture system for intestinal stem cells derived from human intestinal organoids

Ohman Kwon [1], Hana Lee [1], Jaeeun Jung [1], Ye Seul Son [1], Sojeong Jeon [1], Won Dong Yu[1,2], Naeun Son[1,2], Kwang Bo Jung [1], Eunho Choi [1,2], In-Chul Lee [3,4], Hyung-Jun Kwon[3,4], Chuna Kim[1,2,5], Mi-Ok Lee [1,2], Hyun-Soo Cho [1,2,6], Dae Soo Kim [1,2] & Mi-Young Son [1,2,6] ✉

Three-dimensional human intestinal organoids (hIO) are widely used as a platform for biological and biomedical research. However, reproducibility and challenges for large-scale expansion limit their applicability. Here, we establish a human intestinal stem cell (ISC) culture method expanded under feeder-free and fully defined conditions through selective enrichment of ISC populations (ISC[3D-hIO]) within hIO derived from human pluripotent stem cells. The intrinsic self-organisation property of ISC[3D-hIO], combined with air-liquid interface culture in a minimally defined medium, forces ISC[3D-hIO] to differentiate into the intestinal epithelium with cellular diversity, villus-like structure, and barrier integrity. Notably, ISC[3D-hIO] is an ideal cell source for gene editing to study ISC biology and transplantation for intestinal diseases. We demonstrate the intestinal epithelium differentiated from ISC[3D-hIO] as a model system to study severe acute respiratory syndrome coronavirus 2 viral infection. ISC[3D-hIO] culture technology provides a biological tool for use in regenerative medicine and disease modelling.

Three-dimensional (3D) human intestinal organoids (hIOs) can recapitulate the simplified structure and function of the intestine owing to their self-renewal and differentiation capability. Therefore, hIOs have emerged as an in vitro model system for studying human intestinal developmental and functional properties[1,2]. However, large-scale cultivation and its application in high-throughput screening or drug development are limited because of the irregular structure and inefficiency of 3D cultures in Matrigel domes[3,4]. Recently, a high-throughput automated organoid culture technology based on microcavity arrays for the homogenous and large-scale cultivation of hIOs has been developed[5]. Microcavity arrays, however, require a special platform device and are difficult to use.

To overcome these limitations, we aimed to develop a technology for easy and homogenous cultivation of intestinal organotypic models, such as hIOs, in a general laboratory. Standardising culture methods for manipulating intestinal stem cells (ISCs) can provide a reasonable solution for developing homogenous and large-scale intestinal organotypic cultures derived from ISCs. The rapid expansion of ISCs is critical for developing a scalable culture system, as is controlling the number of initial ISCs to improve homogeneity. ISCs, which are rapidly expanded in feeder-free culture conditions, can be the most promising transplantable cell source for intestinal epithelial regeneration[6–8]. In addition, since hIOs comprise only a minor portion (>1%) of stem cells[9], in vitro culturable ISCs can be used as a model system to study ISC

[1]Korea Research Institute of Bioscience and Biotechnology (KRIBB), Daejeon 34141, Republic of Korea. [2]KRIBB School of Bioscience, Korea University of Science and Technology (UST), Daejeon 34113, Republic of Korea. [3]Korea Research Institute of Bioscience and Biotechnology (KRIBB), Jeongeup 56212, Republic of Korea. [4]KRIBB, Korea Preclinical Evaluation Center, Jeongeup 56212, Republic of Korea. [5]KRIBB, Aging Convergence Research Center, Daejeon 34141, Republic of Korea. [6]Department of Biological Science, Sungkyunkwan University, Suwon 16419, Republic of Korea. ✉e-mail: myson@kribb.re.kr

biology and gene editing cell sources for successfully mimicking normal physiology and pathology of the human intestine[10].

In 2015, a two-dimensional (2D) culture technique for ground-state stem cells of the human gastrointestinal tract was reported[11,12]. The ground-state stem cells stably propagated without genome instability and easily and efficiently differentiated into intestinal epithelial cells in an air-liquid interface (ALI) culture. However, the ground-state stem cell culture method has disadvantages. the medium composition is undefined because of the use of conditioned medium and serum, and it must be cultured on mouse 3T3 feeder cells.

In this study, we develop a 2D monolayer culture system for human ISCs derived from 3D-hIO (ISC[3D-hIO]) in a chemically fully defined culture medium under feeder-free conditions. We show that these cells are mainly composed of stem cells and precursor cells and that R-spondin 1 (RSPO1), epidermal growth factor (EGF), and prostaglandin E2 (PGE$_2$) supplementation are essential for cell survival and proliferation. ISC[3D-hIO] monolayers can be cryopreserved and subsequently thawed, allowing for the production of a quality-controlled ISC population. Moreover, ISC[3D-hIO] can be used as a starting cell source for reliable production of a two-and-a-half-dimensional (2.5D) intestinal epithelial model with repetitive protruding villus-like structures through ALI culture. Gene expression patterns of ISC[3D-hIO] closely reflect those of matching original hIO in terms of intestinal maturity. This technology offers a powerful tool for studying human ISC and its potential applications in regenerative medicine and disease modelling.

## Results

### Chemically-defined feeder-free 2D culture system for human ISC

To establish a culture method for obtaining expandable ISCs, we isolated ISC[3D-hIO] from human pluripotent stem cell (hPSC)-derived 3D hIOs (Fig. 1a). To achieve this, single cells and small clumps dissociated from hIOs were seeded on a mitomycin C (MMC)-treated feeder. ISC[3D-hIO] formed small colonies after attaching to feeder cells, and the colony size rapidly increased due to extensive cell proliferation (Fig. 1b and Supplementary Fig. 1a). We tested several types of extracellular matrix (ECM) to replace feeder cells to develop a feeder-free culture system with a chemically fully defined growth medium. Compared to gelatin or collagen type I conditions, ISC[3D-hIO] could grow and proliferate more effectively on the surface coated with 1% Matrigel (Fig. 1c−e). Also, we were able to confirm a 100% cell survival rate without detecting any dead cells and a stable expression of proliferation marker protein such as KI67, when culturing ISC[3D-hIO] on the 1% Matrigel-coated culture dishes (Supplementary Fig. 1b, c).

To optimise the growth medium for ISC[3D-hIO] isolation and expansion, we evaluated combinations of various growth factors and regulators that were previously reported to be involved in the maintenance of human colon crypts in vitro[13,14]. We found that supplementation with RSPO1, EGF, and PGE$_2$ in the culture medium was indispensable for the ISC[3D-hIO] culture (Fig. 2a, b). RSPO1 promotes ISC proliferation in vivo, and hIO growth in vitro and that canonical Wnt signalling is required for ISC function[15,16]. In line with these findings, RSPO1 depletion or WNTi treatment significantly suppressed proliferation of ISC[3D-hIO] (Supplementary Fig. 2a). Further analysis of RSPO1-depleted cells revealed decreased expression of ISC markers (LGR5, CD44, SOX9, LRIG1) and β-catenin target genes (AXIN2, CTNNB) and decreased numbers of KI67/EdU-positive cycling cells (Fig. 2c, d). These findings suggest that RSPO1 is essential for maintaining stemness during ISC[3D-hIO] culture. Next, we examined the effect of EGF depletion on ISC[3D-hIO] cells and found that EGF is required for ISC[3D-hIO] proliferation and survival (Fig. 2a, b, e). EGF depletion from the culture medium induced an extensive decrease in the number of proliferating cells as well as the appearance of dying cells 24 h after EGF depletion or supplementation with the MEK inhibitor PD0325901. In addition, PGE$_2$ depletion appeared to suppress the proliferation of ISC[3D-hIO] cells

(Fig. 2a, b, f). PGE$_2$ can activate downstream signalling pathways through prostaglandin E receptors (PTGER1-4)[17]. Therefore, we investigated the levels of PTGER1-4 expression in ISC[3D-hIO]. PTGER 2 and 4 were highly expressed in ISC[3D-hIO] (Supplementary Fig. 2b), and only the PTGER4 inhibitor inhibited ISC[3D-hIO] cell proliferation (Fig. 2f). Except for RSPO1, EGF, and PGE$_2$, we also evaluated the remaining ingredients to improve the culture efficiency of ISC[3D-hIO]. Most components showed no differences in cell growth immediately after withdrawal (Supplementary Fig. 2c, d, P0). However, all factors affected the survival efficiency of ISC[3D-hIO] cells after passaging (Supplementary Fig. 2c, d, P1). Therefore, we selected ISC[3D-hIO] full growth medium with chemically defined factors for long-term culture (Supplementary Table 1).

The ISC[3D-hIO] was stably maintained in a full growth medium via serial subculture for more than 30 passages and showed high recovery rates and viability (over 90%) after freeze-thawing processes (Fig. 2g and Supplementary Fig. 2e). The ISC[3D-hIO] showed an average expansion rate of 3.01 ± 0.097 fold in 7 days over 10 passages (Supplementary Fig. 2f). To assess the genome stability of ISC[3D-hIO], we performed whole-genome profiling of copy number variation (CNV) using whole genome short-read sequencing data (Supplementary Fig. 3). The genomic stability of ISC[3D-hIO] was well preserved for at least 6 months (P27) without structural variation (Supplementary Fig. 3a). While overall genome of ISC[3D-hIO] remained highly stable, amplified region was found in chromosome 15 at P54 (Supplementary Fig. 3b). For the first 2 days after passaging and thawing, the inclusion of a Rho kinase inhibitor (ROCKi; Y-27632) and Notch activator (Jagged-1 or valproic acid) greatly increased cell viability (Supplementary Fig. 4a). ISCs[3D-hIO] cells were grown in a flat monolayer in 2D culture conditions (Supplementary Fig. 4b), and most of the ISC[3D-hIO] were viable and dead cells were hardly found in LIVE/DEAD assay (Supplementary Fig. 4c).

### Cell composition analysis of ISC[3D-hIO] at single -cell resolution

We performed single-cell RNA sequencing (scRNA-seq) to systematically analyse the cell composition of ISC[3D-hIO]. Our dataset comprised 7,034 high-quality cells, and ISC[3D-hIO] exclusively consisted of intestinal epithelial cells (EPCAM and CDH1) devoid of mesenchymal cell markers (MFAP4, COL1A2, and DCN) (Fig. 3a). The scRNA-seq dataset was then combined with a public dataset of human foetal and adult tissue-derived intestinal epithelia generated by Elmentaite et al. (Supplementary Fig. 5)[18,19]. When the scRNA-seq data was combined, ISC[3D-hIO] matched the foetal small intestine epithelium best during the first trimester (6−8 weeks) (Fig. 3b, c). Furthermore, differentially expressed marker gene analysis revealed nine ISC[3D-hIO] clusters, including 20.5% S phase cells, 5.9% G2/M phase cells, 17% LGR5$^+$ stem cells, 47.9% early enterocyte 1, 7.8% early enterocyte 2, 0.7% enterocytes, and 0.03% goblet progenitors (Fig. 3d, e, and Supplementary Fig. 6a−c). A prominent portion of ISC[3D-hIO] is classified as stem cells and progenitors based on the expression of foetal ISC marker genes (LDHB, EIF3E, SOX9, and SHH) (Fig. 3f, g)[20,21]. Although ISC[3D-hIO] had a small subset of FABP1$^+$ enterocytes, MUC2$^+$ goblet cells and CHGA$^+$ enteroendocrine cells were not identified (Fig. 3g and Supplementary Fig. 6d). These findings suggest that ISC[3D-hIO] mostly comprises stem cells and progenitor cells, with relatively few differentiated cells.

### Differentiation into 2.5D intestinal epithelium using air-liquid interface (ALI) cultures

In vitro intestinal epithelial cultures should resemble the human intestine in vivo for functional analysis and application. ISC[3D-hIO] was differentiated into intestinal epithelium with villus-like protrusion structures using ALI culture on Transwell membrane inserts to establish an in vitro culture model with properties of the human intestine, such as structure and cellular diversity (Fig. 4a). Only five factors (RSPO1, EGF, PGE$_2$, SB202190, and nicotinamide) were present in the designated minimal medium for ALI differentiation when ISC[3D-hIO] were differentiated into 2.5D intestinal epithelium from human embryonic

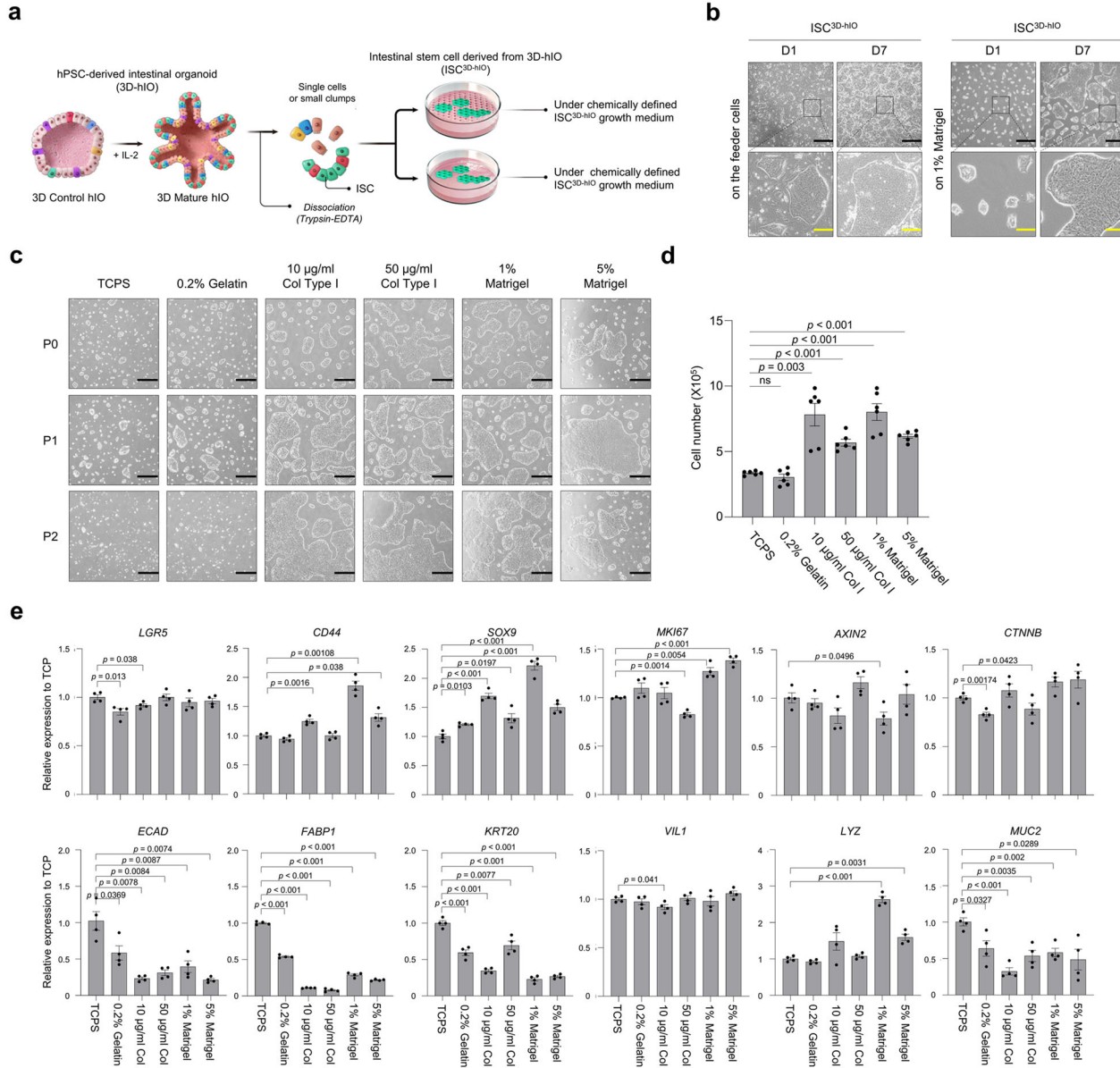

**Fig. 1 | Establishment of 2D ISC³ᴰ⁻ʰᴵᴼ culture method. a** Schematic representation of the developed method. **b** ISC³ᴰ⁻ʰᴵᴼ morphologies on the feeder or 1% Matrigel-coated plate at days 1 and 7. ISCs were generated from hiPSC-derived hIOs (n = 3 samples/group). Black scale bar: 200 μm. Yellow scale bar: 50 μm. **c** Efficiency of cell attachment at P0, P1, and P2, Black scale bar: 200 μm. **d** Average cell number 1 week after cell seeding; Data represents the mean ± SEM (n = 6 biological samples), and a two-tailed t-test was applied to measure p values between the control (TCPS) and the various coating materials. **e** Relative expression of stem cell marker genes (*LGR5, CD44, SOX9, MKI67, AXIN2, and CTNNB*), and differentiated cells (*VIL1, ECAD, FABP1, KRT20, LYZ*, and *MUC2*). Data represents the mean ± SEM (n = 4 biological samples), and a two-tailed t-test was applied to measure p values between the control (TCPS) and the various coating materials.

stem cells (hESCs), patient derived human induced pluripotent stem cells (Patient #1-, and Patient #2-hiPSCs) or EGFP-expressing stable hiPSC line (Fig. 4b, c, and Supplementary Fig. 7a, b). The expression of ISC markers (*LGR5, CD44*, and *MKI67*) rapidly decreased as early as day 4, whereas the expression of the differentiation cell markers (*ECAD, FABP1, KRT20, VIL1, LCT, LYZ*, and *MUC2*) increased continuously (Fig. 4d, e). The epithelium derived from ISC³ᴰ⁻ʰᴵᴼ developed villus-like serpentine patterns[11], and H&E-stained sections showed that the thickness of villi-like folds increased significantly with the culture period in ALI (Fig. 4e, f). Consistently, the Measurement of transepithelial electrical resistance (TEER) values, representing epithelial integrity, also gradually increased to 209 ± 12.25 ohm*cm² after 12 days of ALI culture (Fig. 4g).

Next, we analysed RNA-seq data generated from ISC³ᴰ⁻ʰᴵᴼ and ALI-differentiated cells from ISC³ᴰ⁻ʰᴵᴼ to elucidate the mechanism by which intestinal epithelial monolayers change after differentiation. Principal component analysis (PCA) of differentially expressed genes (DEGs) of ISC³ᴰ⁻ᶜᵒⁿᵗʳᵒˡ ʰᴵᴼ and ISC³ᴰ⁻ᵐᵃᵗᵘʳᵉ ʰᴵᴼ derived from foetal-like control[22] and in vitro mature 3D hIO induced by IL-2 treatment[23], respectively, revealed that the ALI-differentiated cells from each showed distinct status when compared to hPSC, control and in vitro mature 3D hIO, functional intestinal epithelial cells (hIECs)[24], and human small intestine tissue (hSI) (Fig. 4h). Although the global gene expression profiles revealed considerable gaps between ALI-differentiated cells and hSI when compared to 3D mature hIO and functional hIECs, functional annotation clustering demonstrated that several biological processes were

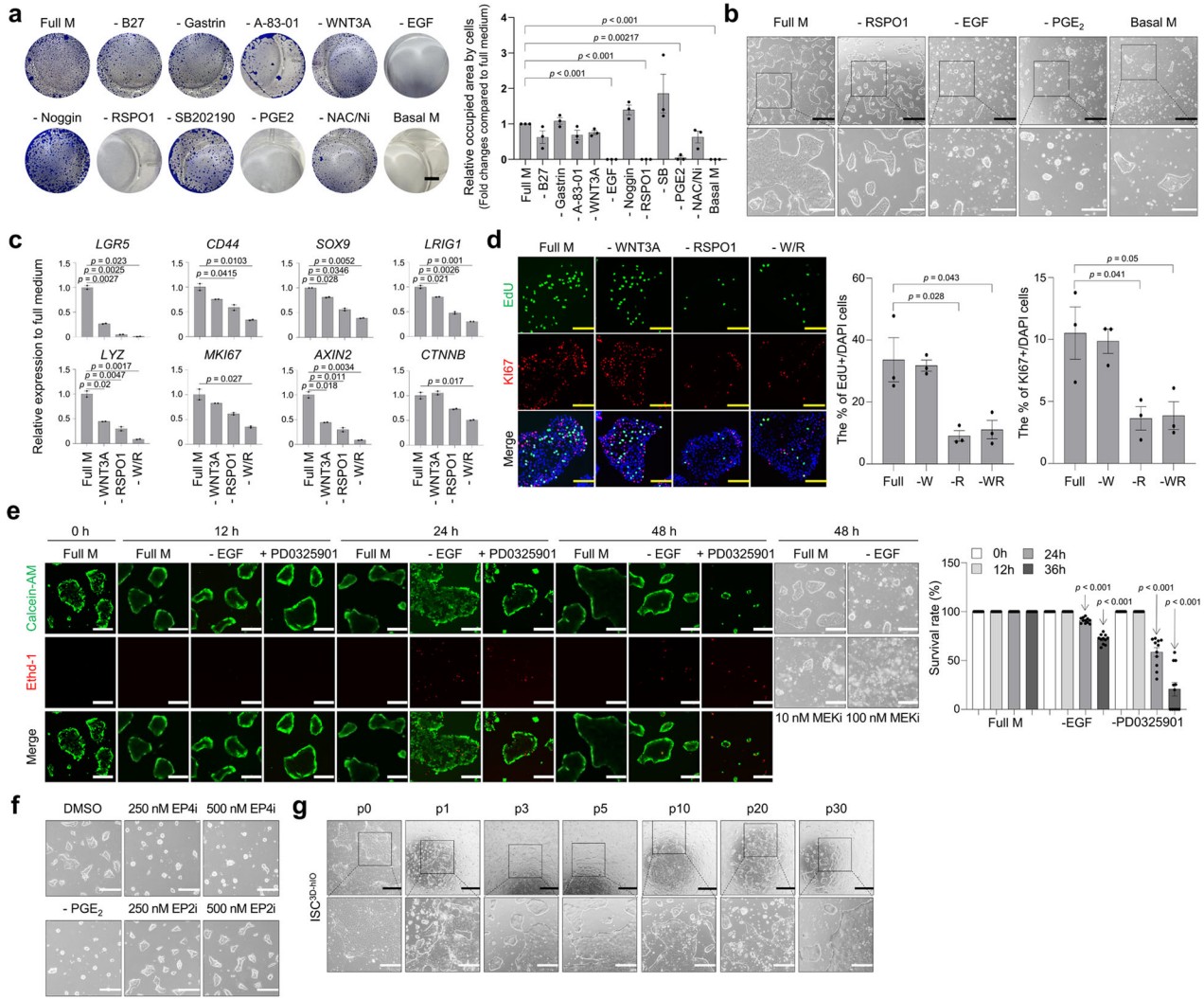

**Fig. 2 | Optimisation of ISC^{3D-hIO} culture media. a** Representative images of crystal violet (CV) stained ISC^{3D-hIO} colonies with depletion of single component from growth media and quantification of occupied area by ISC^{3D-hIO}. Black scale bar, 50 mm. Data represent the mean ± SEM (n = 3 biological samples), and a two-tailed t-test was applied to measure p values between the control cells (Full M) and the cells grown in the various conditions. **b** Representative images of ISC^{3D-hIO} grown in full growth medium (Full M), depletion of RSPO1, EGF, or PGE2, or basal medium (Basal M) (n = 3 samples/group). Black scale bar, 200 μm. White scale bar, 100 μm. **c** Relative expression of marker genes in ISC^{3D-hIO} grown in Full M, depletion of WNT3A, RSPO1, or WNT3A/RSPO1. Data represent the mean ± SEM (n = 2 biological samples), and a two-tailed t-test was applied to measure p values between the control cells (Full M) and the cells grown in the growth factor depleted conditions. **d** Immunofluorescence images and quantification analysis in ISC^{3D-hIO} grown in Full M, depletion of WNT3A, RSPO1, or WNT3A/RSPO1. Yellow scale bar: 50 μm. Data represent the mean ± SD (n = 3 biological samples), and a two-tailed t-test was applied to measure p values between the control cells (Full M) and the cells grown in the growth factor depleted conditions. **e** Live (Calcein-AM)/Dead (EthD-1) analysis of ISC^{3D-hIO} in Full medium or after depletion of EGF, or treatment with PD0325901 at 0, 12, 24, and 48 h, and representative images of ISC^{3D-hIO} grown in Full M, depletion of EGF, treatment with 10 nM PD0325901, or treatment with 100 nM PD0325901. White scale bar: 100 μm. Data represent the mean ± SD (n = 4 biological samples, each with 2–3 technical replicates), and a two-tailed t-test was applied to measure p values. **f** Representative images of ISC^{3D-hIO} grown in treatment with DMSO, treatment with 250 nM EP2i, treatment with 500 nM EP2i, treatment with 250 nM EP4i, treatment with 500 nM EP4i, or depletion of PGE2 (n = 3 samples/group). White scale bar: 100 μm. **g** Morphologies of ISC^{3D-hIO} at passages 0, 1, 3, 5, 10, 20, and 30 (n = 3 samples/group). Black scale bar: 200 μm. White scale bar: 100 μm.

differently regulated between ISC^{3D-hIO} and ALI-differentiated cells (Fig. 4i). In particular, DNA replication, DNA biosynthesis, and G1/S cell cycle transition were downregulated in ALI-differentiated cells, but other biological processes related to intestinal epithelial development and maturation, such as oestrogen biosynthesis (*AKR1B15, DHRS11*), O-glycan processing (*GALNT4, GALNT5*), retinol metabolism (*DHRS3, RDH10*), xenobiotic metabolism (*AADAC, NR1I2*), steroid metabolism (*SULTE1, NR1I2*), oxidant detoxification (*DUOX2, FABP1*), amino acid uptake (*SLC6A20, SLC43A1*), and maintenance of permeability (*CLDN3*) were significantly upregulated in ALI-differentiated cells (Fig. 4j). These findings suggest that under ALI conditions, ISC^{3D-hIO} has the capacity to differentiate into a functional intestinal epithelium.

## Application of ISC^{3D-hIO} for gene editing and transplantation

ISC^{3D-hIO} grows rapidly and is cultured under controlled conditions, making it suitable for use in various application studies. First, we used lentiviral gene transduction to create genetically modified ISC^{3D-hIO} lines. Single-cell dissociated ISC^{3D-hIO} was transduced by lentiviral spin infection and plated on a 1% Matrigel-coated culture plate for stable gene transduction (Fig. 5a). ISCs^{3D-hIO}-expressing GFP appeared partially immediately after lentiviral infection, but only GFP-expressing ISCs^{3D-hIO} remained after puromycin selection (Fig. 5b, upper panels). To establish isogenic ISCs^{3D-hIO} populations, GFP-expressing ISCs^{3D-hIO} were dissociated and filtered into single cells before being sparsely re-plated on the 1% Matrigel-coated culture plate (Fig. 5b, left bottom

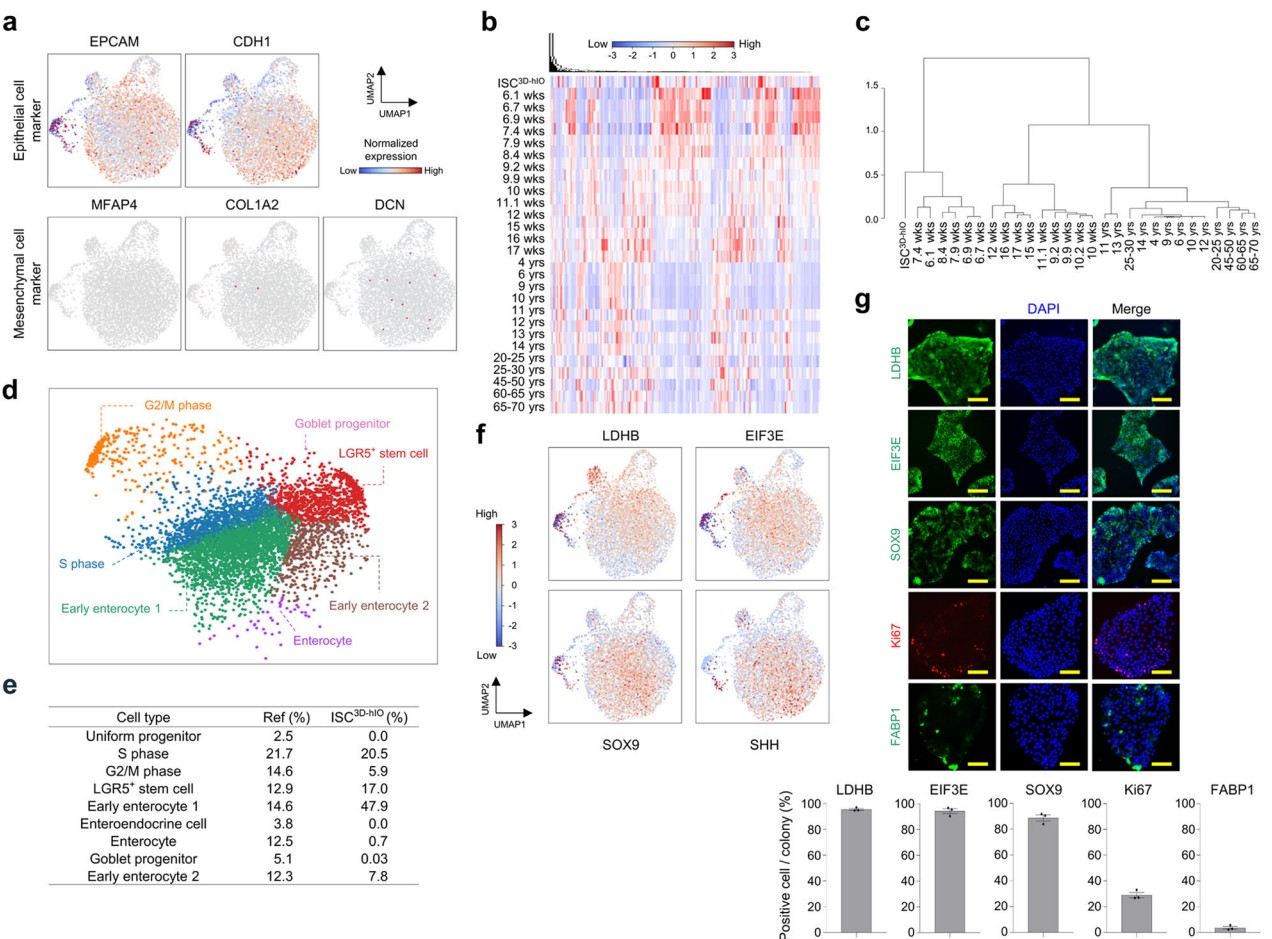

**Fig. 3 | scRNA-seq analysis for characterising ISC$^{3D\text{-}hIO}$ composition. a** Feature plot of epithelial cell markers (*EPCAM* and *CDH1*) and mesenchymal cell markers (*MFAP4, COL1A2* and *DCN*). Heatmap (**b**) and dendrogram (**c**) depict molecular transition as age increases. **d** UMAP plot with single cells of ISC3d-hIO coloured by cell types. **e** List of cell types within ISC$^{3D\text{-}hIO}$ and their percentages from scRNA-seq data. **f** Feature plot of the foetal intestinal stem cell markers (*LDHB, EIF3E, SOX9, and SHH*). **g** Immunofluorescence images of foetal intestinal stem cell markers (*LDHB, EIF3E, SOX9, and KI67*) and enterocyte marker (*FABP1*). Quantification of LDHB, EIF3E, SOX9, Ki67 and FABP1 positive cells in ISC$^{3D\text{-}hIO}$ colonies Data represent the mean ± SD (*n* = 3 biological samples). Yellow scale bar: 50 µm.

panel). Single colonies were enzymatically detached after isogenic ISC$^{3D\text{-}hIO}$ expansion (Fig. 5b, right bottom panel) using collagenase IV and dispase treatment. Pedigree lines of ISCs$^{3D\text{-}hIO}$ were grown in our culture conditions for several days (Fig. 5c) and differentiated into 3D hIOs and 2.5D intestinal epithelium via ALI cultures (Fig. 5d, e). These results suggest that ISCs$^{3D\text{-}hIO}$ can be applied to genome editing strategies to construct stable cell lines.

Following that, we transplanted ISCs$^{3D\text{-}hIO}$ into the EDTA-injured colonic epithelium of immunodeficient NOD/SCID deleted IL2Rg gene (NIG) mice and observed them for 2 weeks to evaluate their regenerative potential (Fig. 6a). The weight change after rectal ISC$^{3D\text{-}hIO}$ transplantation was significantly higher on day 14 in the colonic epithelial injury model compared with that in the matrigel transplant group (Fig. 6b). The degree of recovery of the damaged colonic epithelium was confirmed by directly observing the intestinal environment via colonoscopy at 0, 3, and 14 days after transplantation (Fig. 6c and Supplementary Fig. 8a–c). On day 3, both the Matrigel and ISC$^{3D\text{-}hIO}$ transplant groups showed inflammation due to intestinal epithelial damage caused by EDTA-induced injury (Fig. 6c). Fourteen days after transplantation, the ISC$^{3D\text{-}hIO}$ transplant group had less epithelial damage than that in the Matrigel transplant group (Fig. 6c). The Matrigel transplant group also had a 50% mortality rate, which was higher than that in the ISC$^{3D\text{-}hIO}$ transplant group (20%) (Supplementary Fig. 8d). In addition, ISCs$^{3D\text{-}hIO}$ labelled with a near-infrared fluorescent lipophilic dye (DiR) were transplanted into EDTA-damaged colonic

epithelium to confirm the actual transplanted cells. Two weeks after transplantation, DiR-labelled ISCs$^{3D\text{-}hIO}$ showed a clear fluorescence signal in colonic tissue ex vivo (Fig. 6d). Furthermore, 2 weeks after the eGFP labelled reporter ISC (ISCs$^{3D\text{-}ISX\text{-}eGFP\text{-}hIO}$), the implantation site with GFP fluorescence expression could be found near the rectum using a fluorescent stereoscope (Fig. 6e). Histological analysis with H&E and AB-PAS staining showed that xenografted ISCs$^{3D\text{-}hIO}$ successfully regenerated colonic epithelium (Fig. 6f). The crypt structure of the ISC$^{3D\text{-}hIO}$ implantation site was actively formed, and mucin secretion was acquired. Moreover, immunofluorescence staining with human-specific cytokeratin (hCytokeratin) confirmed the xenografted ISCs$^{3D\text{-}hIO}$, and a strong signal, particularly near the rectum (Fig. 6g). The fluorescence intensities of hCytokeratin were significantly increased in ISC$^{3D\text{-}hIO}$ transplantation group compared to those in Matrigel group (Fig. 6g). Thus, in vivo data indicated that ISC$^{3D\text{-}hIO}$ is an implantable source of tissue regeneration capable of forming intestinal epithelial structures.

## Modelling of severe acute respiratory syndrome coronavirus 2 (SARS-CoV-2) infection in ALI-differentiated intestinal epithelium

The 2.5D intestinal epithelium produced by ALI culture is a suitable model for recapitulating key aspects of host-pathogen interaction. Since the SARS-CoV-2 virus has previously been shown to directly infect enterocytes in 3D hIOs[25], we attempted to generate a SARS-CoV-

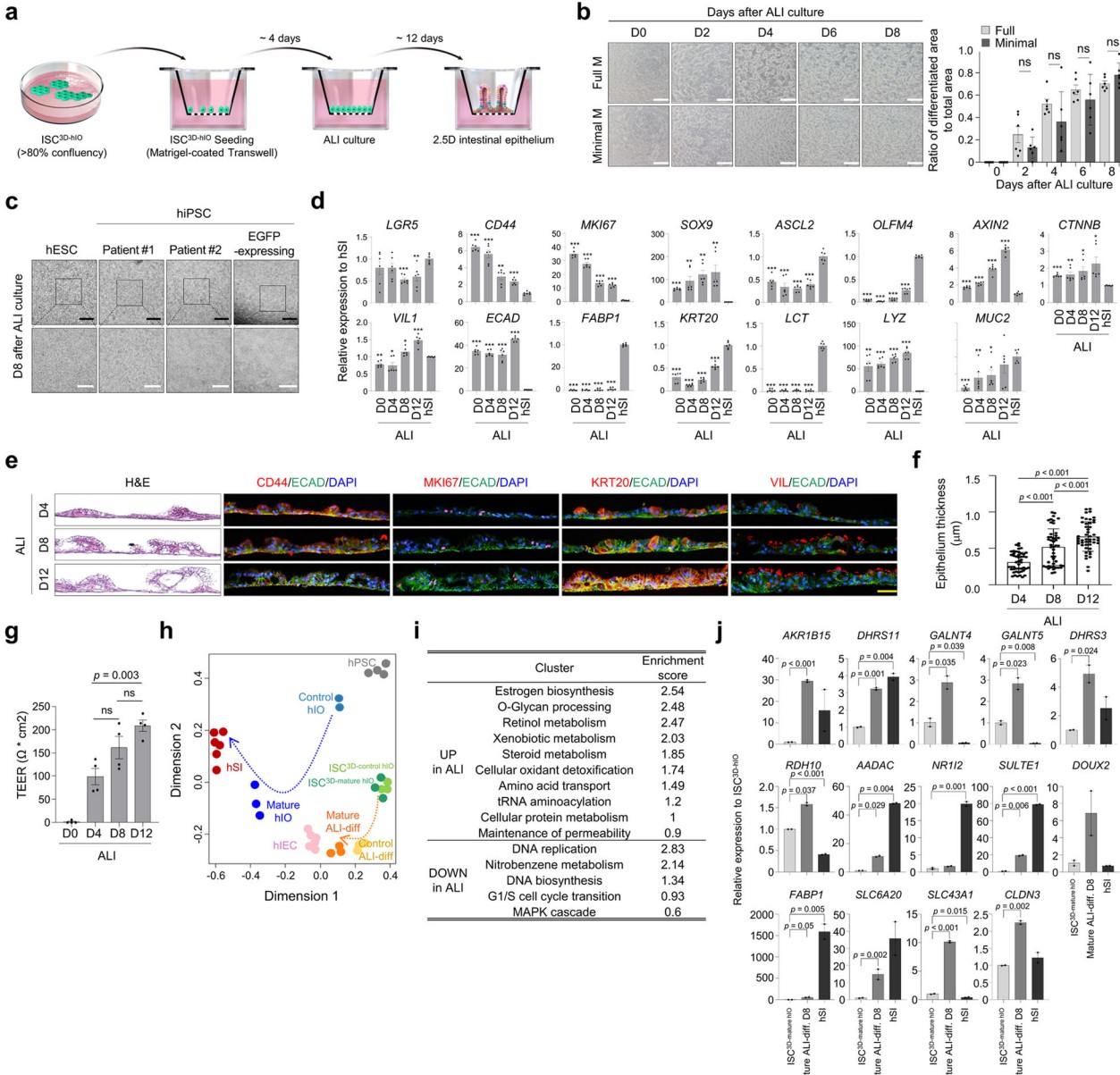

**Fig. 4 | Differentiation of ISC^3D-hIO into intestinal epithelium via ALI. a** Schematic representation of the differentiation method. **b** Representative images of 2.5D intestinal epithelium at days 0, 2, 4, 6, and 8 after ALI culture in Full M or the defined Minimal M. White scale bar: 100 μm. Quantification analysis of 2.5D intestinal epithelium in Full M or Minimal M. Data represent the mean ± SD ($n = 3$ biological samples, each with 2 technical replicates), and a two-tailed $t$-test was applied to measure $p$ values between the control cells in the Full M and the cells grown in the Minimal M. **c** 2D intestinal epithelium morphologies derived from hESC and hiPSCs (patient #1, patient #2, and genome edited) at day 8 after ALI culture ($n = 3$ samples/group). Black scale bar: 200 μm. White scale bar: 100 μm. **d** Relative expression of stem cell and differentiated cell marker genes in ALI-differentiated cells at days 0, 4, 8, and 12 after air exposure and hSI. Data represent the mean ± SD ($n = 3$ biological samples, each with 2 technical replicates), and a two-tailed $t$-test was applied to measure $p$ values between the human intestinal tissue (hSI) and the ALI-differentiated cells. The exact $p$ values represented in the source data. H&E (**e**, left) and immunofluorescence staining (**e**, right) of intestinal markers and epithelium

thickness at days 4, 8, and 12 after ALI culture. **f** Yellow scale bar: 100 μm. Data represent the mean ± SD ($n = 48$ biological samples), and a two-tailed $t$-test was applied to measure $p$ values among the groups. **g** TEER values of ALI-differentiated cells at days 0, 4, 8, and 12 after air exposure. Data represent the mean ± SD ($n = 4$ biological samples), and a two-tailed $t$-test was applied to measure $p$ values between the control (D0) and days after ALI-differentiation. **h** MDS plot shows the pairwise distance between samples. Nine homogeneous sample groups were observed: hPSC (grey, $n = 4$), P0 3D hIO (light blue, $n = 2$), mature 3D hIO (blue, $n = 3$), functional hIECs (pink, $n = 6$), immature ISC^3D-hIO (green, $n = 3$), mature ISC^3D-hIO (dark green, $n = 3$), immature ALI (yellow, $n = 3$), mature ALI (orange, $n = 3$), and hSI (red, $n = 6$). **i** Enriched functional clusters of biological processes (BP) and enrichment scores encompassing DEGs between ISC^3D-hIO and ALI-differentiated cells. **j** Relative expression of genes up-regulated in ALI-differentiated cells compared to ISC^3D-hIO. Data represent the mean ± SEM ($n = 2$ biological samples), and a two-tailed $t$-test was applied to measure $p$ values between control cell (ISC^3D-hIO) and differentiated cells (Mature ALI-diff. D8 and hSI).

2 virus infection model using 2.5D intestinal epithelium. We generated 2.5D intestinal epithelium using ISC^3D-control hIO and ISC^3D-mature hIO derived from foetal-like control hIO[22,26] and adult-like mature hIO[23,27], respectively (Fig. 7a, b). Each ISC^3D-hIO retained its origin hIO's intestinal maturity-specific gene expression profiles, including *DPP4*, *DEFA5*,

*OLFM4*, and *MUC2*[23,26,28] (Fig. 7c). Notably, angiotensin-converting enzyme 2 (*ACE2*) expression was increased in ALI-differentiated intestinal epithelium from ISC^3D-mature hIO, whereas the expression level of the type II transmembrane serine protease (*TMPRSS2*) and the disintegrin and metalloproteinase 17 (*ADAM17*) was not significantly

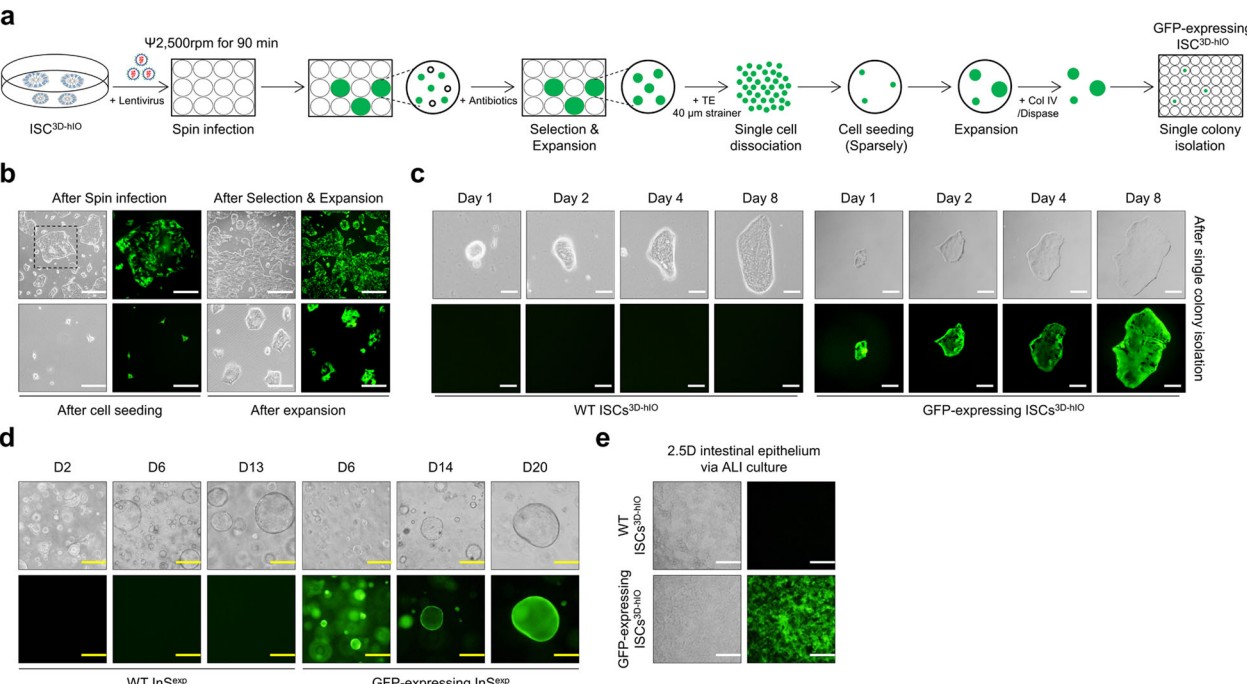

**Fig. 5 | Generation of an eGFP-expressing ISC³ᴰ⁻ʰᴵᴼ reporter cell line. a** Schematic overview of the methodology. eGFP expression by lentiviral infection is represented by green. **b** Representative images of ISC³ᴰ⁻ʰᴵᴼ after infection, selection & expansion, low-density cell seeding, and expansion to form colonies. White scale bar: 100 μm. Cell growth images of the single colony (**c**) and 3D expandable intestinal spheres (InSᵉˣᵖ) (**d**) are grown from wild-type or single EGFP-expressing cells (n = 3 samples/group). White scale bar: 250 μm. Yellow scale bar: 50 μm. **e** 2.5D intestinal epithelium via ALI differentiation (n = 3 samples/group). White scale bar: 100 μm.

different from that observed in the bulk RNA-seq data (Supplementary Fig. 9). The increased ACE2 expression in the ALI-differentiated intestinal epithelium from ISC³ᴰ⁻ᵐᵃᵗᵘʳᵉ ʰᴵᴼ was also confirmed at the transcriptional and translational levels (Fig. 7d, e). After 24 h of SARS-CoV-2 infection, virus entry and infection were assessed via viral RNA detection, such as the N, E, and RdRP genes. The amount of virus inoculated increased viral infection, and the intestinal epithelium from ISC³ᴰ⁻ᵐᵃᵗᵘʳᵉ ʰᴵᴼ was more susceptible to viral infection than the intestinal epithelium from ISC³ᴰ⁻ᶜᵒⁿᵗʳᵒˡ ʰᴵᴼ (Fig. 7e). On the contrary, the infection of the virus was significantly inhibited when treated with chemical drugs that can disrupt the interaction between ACE2 and SARS-CoV-2 (Fig. 7f)[29,30].

## Discussion

In this study, we attempted to establish a homogenous and stably expandable ISC culture system and subsequently differentiate it into 2.5D intestinal epithelium to develop a highly reproducible and applicable high-throughput screening in vitro intestinal model. The 2D ISC³ᴰ⁻ʰᴵᴼ culture system supports an enriched population of ISCs and early progenitors under fully defined culture media and feeder-free conditions, rapid propagation, long-term maintenance with simple passaging, efficient cryopreservation of ISC³ᴰ⁻ʰᴵᴼ with multiple cycles of freezing and thawing, and highly reproducible differentiation into functional cells when needed. Thus, the ISC³ᴰ⁻ʰᴵᴼ is a desirable cell source for applications such as in vitro model systems for mimicking intestinal physiology, disease modelling, genome editing, and regenerative medicine via cell transplantation.

To mimic in vivo intestinal physiology, these stem cells must be differentiated into intestinal epithelial cells, such as absorptive and secretory cells. Although 3D hIOs are a well-known model system containing differentiated cells, an intestinal epithelial model system capable of quantitative assessment is required to overcome the limitations of 3D hIOs, they cannot be used for conventional intestinal

assays using 2D monolayer cultures[31]. Our method of ALI culture to differentiate ISC³ᴰ⁻ʰᴵᴼ into 2.5D intestinal epithelium provides a controllable in vitro intestinal model with easy access to the lumen, reduced batch variation, and compatibility with functional assays. Evidently, it was observed that a villus-like structure grew from a flat monolayer over time, and the expression levels of differentiated cell marker genes and barrier function also increased during morphogenesis. Furthermore, the 2.5D intestinal epithelium model system enables easy manipulation of intestinal stem cells without the need for the process of stem cell sorting using cell sorting technologies such as FACS. These advantages allow it to be easily integrated with bioengineering technologies such as micropatterned plates and gut-on-a-chip, not just transwell plates.

Based on these advantages, SARS-CoV-2 infection was confirmed, and viral RNAs were expressed in ALI-differentiated intestinal epithelium from ISC³ᴰ⁻ʰᴵᴼ, a model host cell system for SARS-CoV-2 infection. Because *ACE2*, the receptor for SARS-CoV-2, is highly expressed in ALI-differentiated intestinal epithelium from ISC³ᴰ⁻ᵐᵃᵗᵘʳᵉ ʰᴵᴼ compared with that in ALI-differentiated intestinal epithelium from ISC³ᴰ⁻ᶜᵒⁿᵗʳᵒˡ ʰᴵᴼ, SARS-CoV-2 infection occurred more frequently in ALI-differentiated intestinal epithelium from ISC³ᴰ⁻ᵐᵃᵗᵘʳᵉ ʰᴵᴼ. ACE2 plays a key role as a viral receptor for SARS-CoV-2, and TMPRSS2-mediated proteolytic cleavage of the ACE2 cytoplasmic tail increases endosomal internalisation[32,33]. Consistently, we also confirmed the essential role of ACE2 as a viral receptor in ALI-human intestinal epithelium from ISC³ᴰ⁻ʰᴵᴼ, because the reduction of viral infection was observed when treating with chemical inhibitors that inhibit plasma membrane expression of ACE2 or chemical inhibitors that interfere with the binding between SARS-CoV-2 and ACE2[29,30]. Therefore, the sensitivity to SARS-CoV-2 infection in human intestinal epithelium is determined by the expression level of the ACE2, and it can be verified that SARS-CoV-2 is less likely to infect children[34,35] who have less developed SARS-CoV-2 binding sites of ACE2[36,37]. Our data demonstrate that ISC³ᴰ⁻ʰᴵᴼ preserves maturity-

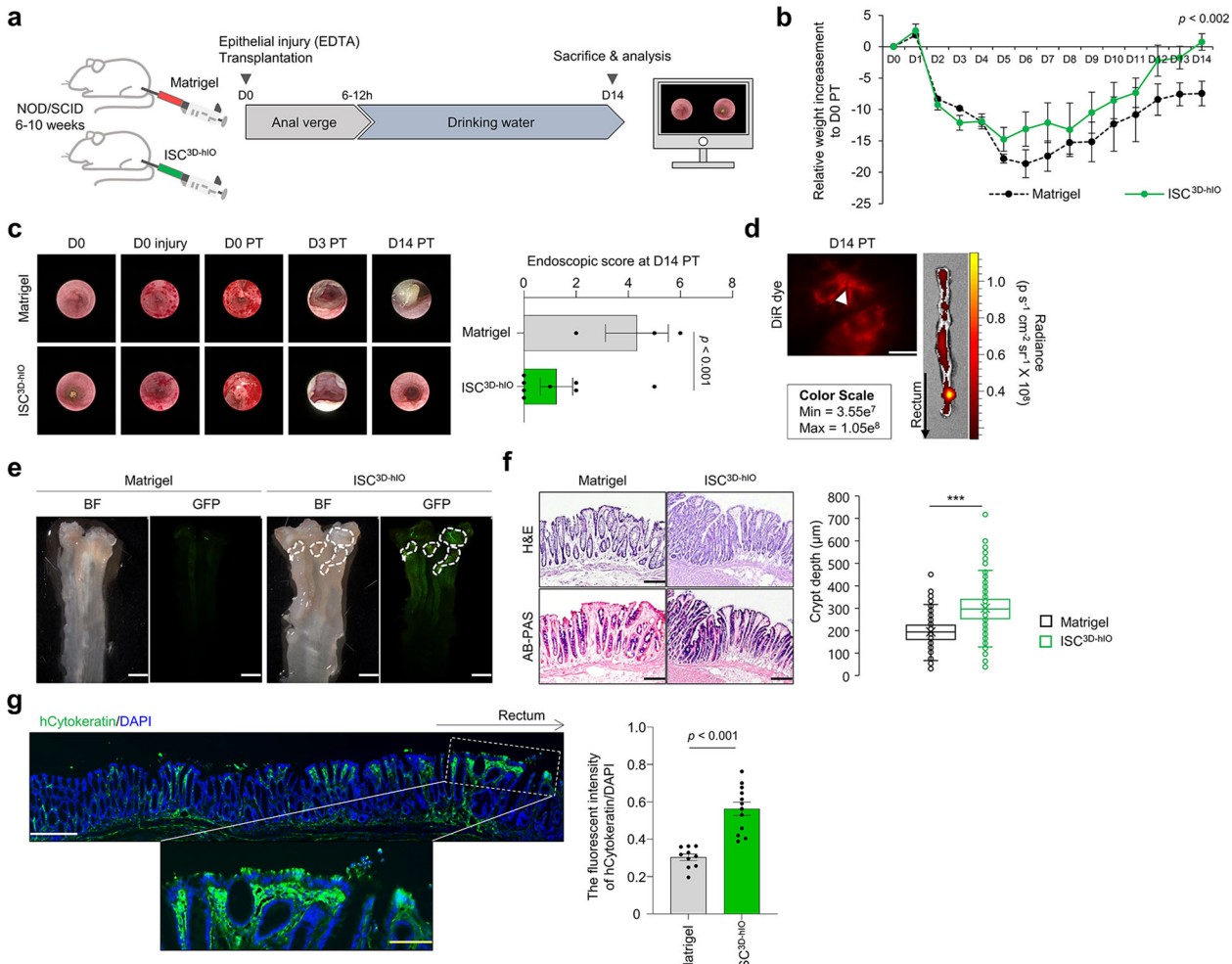

**Fig. 6 | Xenotransplantation of human ISC[3D-hIO] into an EDTA-induced epithelial injury mouse model. a** Schematic representation of the transplantation. **b** Relative weight changes of xenotransplantation for 14 days. Matrigel group (n = 3), ISC[3D-hIO] group (n = 8). Data represent the mean ± SD, and a two-tailed *t*-test was applied to measure *p* values. **c** Colonoscope observation of the mouse colon at day 0, 3, and 14 post-transplantation (PT), and Endoscopic score of the transplanted mice. Matrigel group (n = 3), ISC[3D-hIO] group (n = 8), and a two-tailed non-parametric Mann–Whitney *U* test was applied to measure *p* values. **d** The fluorescent image shows the DiR⁺ grafts 14 days PT (left). White scale bar: 125 μm. IVIS image of the recipient's colon contains DiR⁺ grafts 14 days PT (right) (n = 2 samples). **e** The bright fields of the recipient's colon and fluorescent images of ISCs[3D-ISX-eGFP-hIO] grafts on the colon. Matrigel group (n = 2), ISC[3D-hIO] group (n = 3). White scale bar: 2 mm. **f** Histological analysis of the xenograft tissues (H&E staining, upper) and histopathology of the xenograft colon (AB-PAS, bottom). Black scale bar: 200 μm. The box and scatterplots of crypt depths of Matrigel (n = 675 crypts from three mice) and ISC[3D-hIO] (n = 705 crypts from eight mice) transplanted mouse tissues (H&E staining). The quartiles of the boxplot are mean ± SD, and a Welch's unpaired t-test was applied to measure *p* values. **g** Immunofluorescence images of the recipient's colon with indicated hCytokerat in. White scale bar: 275 μm. Yellow scale bar: 100 μm. Fluorescence intensity of hCytokeratin/DAPI (n of fields = 10 in Matrigel group, n of fields = 12 in ISC[3D-hIO] group). Data represent the mean ± SEM, a two-tailed *t*-test was applied to measure *p* values.

specific gene expression patterns reflecting their hIO origin, such as foetal-like and adult-like maturation characteristics. In line with these findings, the adjustable differentiation of ISC[3D-hIO] derived from hIOs representing a particular state in an intestinal epithelial monolayer can be used for various applications, including intestinal morphogenesis and disease modelling.

Furthermore, when the global gene expression pattern was analysed using bulk RNA-seq, in vivo intestinal metabolism and nutrient transport-related terms were highly enriched in the ALI-differentiated intestinal epithelium than in the ISC[3D-hIO]. However, there are limitations regarding setting and use as a more physiologically mimetic intestine model because the differentiated cells in the 2.5D intestinal epithelium from ISC[3D-hIO] have yet to reach maturity. Indeed, we confirmed that the maturity and functionality of the 2.5D intestinal epithelium from ISC[3D-hIO] were lower than those of the 2D functional intestinal epithelium directly differentiated from hPSCs, as recently reported by our group[24]. These phenomena were presumed to be

related to the abundance of growth factors or cytokines, which maintain stemness; now we are conducting further studies to find the best conditions to enhance the maturity of the 2.5D intestinal epithelium. Furthermore, it is necessary to develop a co-culture system using various stromal cells that can closely mimic the native environment of the human intestine in future studies[38,39].

In this study, we demonstrated the possibility of using ISC[3D-hIO] as an obvious target for genetic engineering and as a transplant source for intestinal diseases. Despite the need for additional functional characterisation and application studies, we suggest that ISC[3D-hIO] is easy to handle and a suitable cell source for various applications, including genetic engineering. We generated eGFP reporter ISC[3D-hIO] cell lines that can efficiently differentiate into hIOs and intestinal epithelium while retaining their morphology. hIOs have mostly been used in previous studies for in vivo transplantation for damaged tissue regeneration[6-8]. These studies suggest that LGR5⁺ stem cells play a key role in cell engraftment and tissue regeneration and can be used for

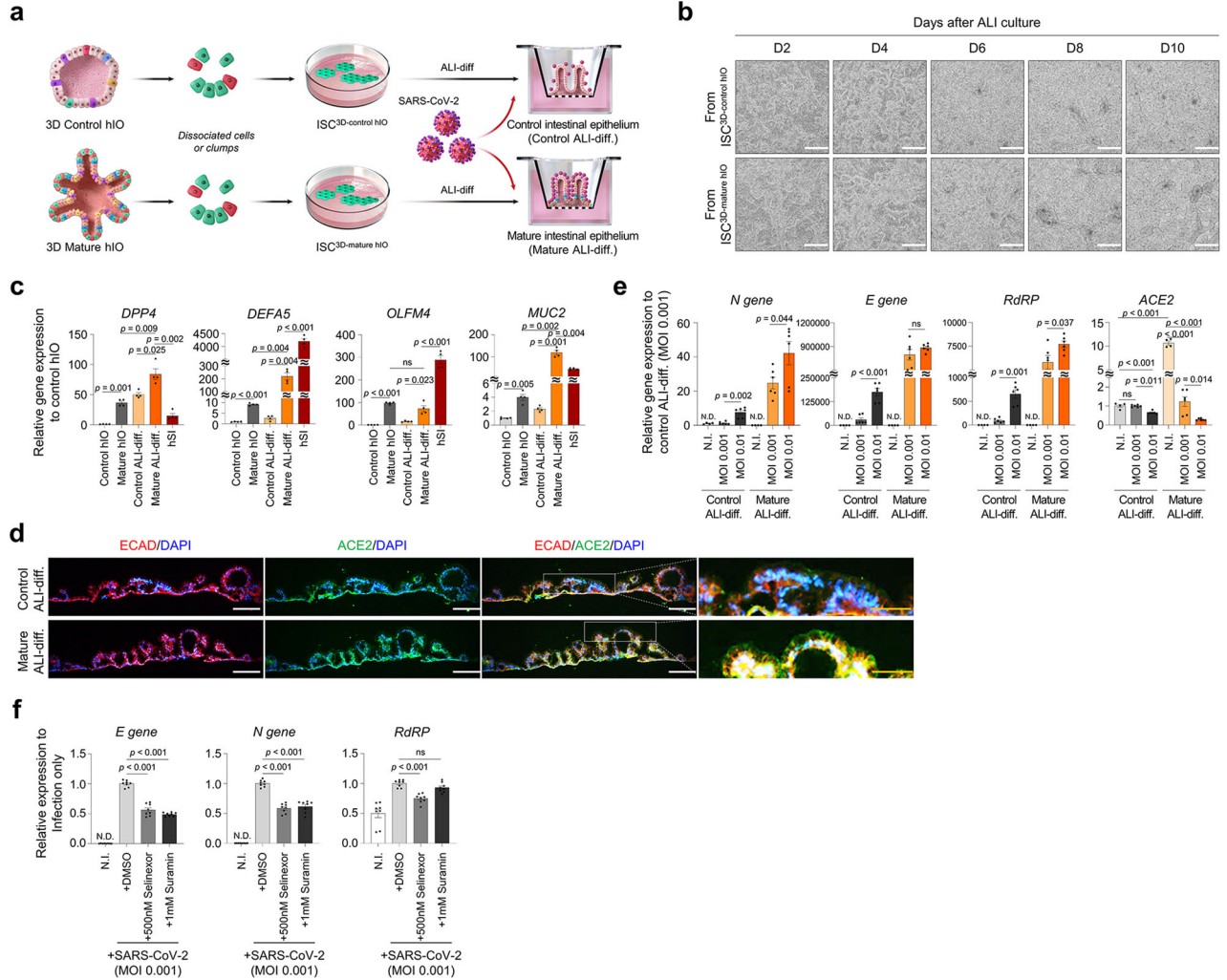

**Fig. 7 | SARS-CoV-2 infection on ALI-differentiated cells differentiated from ISC[3D-control hIO] and ISC[3D-mature hIO]. a** Schematic representation of in vitro SARS-CoV-2 infection model system. **b** Representative images of intestinal epithelium differentiated from either ISC[3D-control hIO] derived from foetal-like control hIO or ISC[3D-mature hIO] derived from adult-like mature hIO at days 2, 4, 6, 8, and 10 after air exposure. White scale bar: 100 μm. **c** Relative expression of intestinal maturation marker genes in control hIO (foetal-like), mature hIO (adult-like), ALI-differentiated cells from ISC[3D-control hIO] (Control ALI-diff.) and ISC[3D-mature hIO] (Mature ALI-diff.), and human small intestine tissue (hSI). Data represent the mean ± SEM ($n = 4$ biological samples), and a two-tailed $t$-test was applied to measure $p$ values for control hIO vs mature hIO, control ALI-diff. vs mature ALI-diff., mature hIO vs mature ALI-diff., and

mature ALI-diff. vs hSI. **d** Immunofluorescence staining of ACE2 in Control ALI-diff. and Mature ALI-diff. White scale bar: 100 μm. Yellow scale bar: 50 μm. **e** Relative expression of SARS-CoV-2 viral genes (N gene, E gene, RdRP) and ACE2 by viral infection in Control ALI-diff. and Mature ALI-diff. Data represent the mean ± SEM ($n = 4$ biological samples for N. I., $n = 6$ biological samples for the other conditions), and a two-tailed $t$-test was applied to measure $p$ values. **f** Perturbation effects by selinexor or suramin on the interaction between SARS-CoV-2 and ACE2 in 2.5D intestinal epithelium. The data represent the mean ± SD ($n = 4$ biological samples, each with 2 technical replicates), and a two-tailed $t$-test was applied to measure $p$ values between control cell (DMSO) and the cells treated with chemicals.

stem cell therapy in regenerative medicine. However, isolating LGR5[+] stem cells from 3D hIO is technically difficult; most cases have transplanted whole organoids or mechanically decomposed organoid fragments[6,7,23,28]. ISC[3D-hIO] is primarily composed of ISCs and early progenitors and can be transplanted without needing a separate process. Transplanted ISC[3D-hIO] was rapidly engrafted into the damaged region and successfully reconstituted the intestinal epithelium within 2 weeks. Furthermore, xenotransplant recipient mice recovered their body weight faster than Matrigel-transplanted control mice, implying a beneficial effect of ISC[3D-hIO] transplantation. Although more research into transplantation methods and long-term monitoring is needed, the current study suggests that ISC[3D-hIO] transplantation is a promising treatment method for patients with gastrointestinal epithelial disorders.

In conclusion, we established an ISC[3D-hIO] culture system that could be highly enriched for rapidly expanding ISCs and early progenitors

derived from 3D hIOs. Our study suggests the feasibility of the ISC[3D-hIO] culture system for applied research, including genetic engineering, regenerative medicine, and disease modelling. It is also conceivable that, with further refinement, the differentiation method into 2.5D intestinal epithelium using ALI Transwell will be compatible with studies such as drug absorption, drug toxicity, and microbe-epithelium interaction analysis.

## Methods

### Ethical considerations of working with human cells and animals

Our research complies with all relevant ethical regulations. All studies based on human pluripotent stem cells were approved by the Korean Public IRB (IRB numbers: P01-201409-ES-01-09, P01-201609-31-002). Six-to–twelve weeks old male NIG mice (NOD/SCID deleted IL2Rg gene) obtained from GHBio Korea were maintained under specific-pathogen-free condition. All animal experiments were approved by the

Institutional Animal Care and Use Committee (IACUC) of KRIBB (approval number: KRIBB-AEC-21236).

## Human pluripotent stem cells

The H9 hESC line was purchased from WiCell Research Institute (Madison, WI, USA). The generation of hiPSC and culture methods of hESC and hiPSCs have been described in a previous report[40]. Briefly, hESCs and iPSCs were maintained on Matrigel (BD Biosciences) in mTesR1 medium without feeders, and routinely passaged every week.

## Differentiation of hPSC into hIO

The hIO differentiation method has previously been described in detail[22,23]. hPSCs were differentiated into definitive endoderm (DE) by treatment with 100 ng/ml activin A (R&D Systems, Minneapolis, MN, USA) for 3 days in RPMI 1640 medium with increasing concentrations of 0%, 0.2%, and 2% defined foetal bovine serum (FBS; Gibco, Cat. No. 16000044, Thermo Fisher Scientific Inc., Waltham, MA, USA). DE cells were then treated for 4 days with RPMI 1640 medium containing 2% dFBS, 500 ng/ml FGF4 (Peprotech, Cat. No.100-31-500, Thermo Fisher Scientific Inc., Waltham, MA, USA), and 500 ng/ml WNT3A (R&D Systems, Cat. No.5036-WN-500, R&D Systems, Minneapolis, MN, USA) to promote differentiation into 3D hindgut spheroids. The spheroids were harvested and embedded in Matrigel (Matrigel® Basement Membrane Matrix, LDEV-free, Cat. No. 354234, Corning, NY, USA), cultured in hIO medium composed of advanced Dulbecco's Modified Eagle Medium (DMEM)/F-12 medium (Gibco, Cat No. 11330-099, Thermo Fisher Scientific Inc., Waltham, MA, USA) containing 1× B27 (Invitrogen, Cat No. 12587-010, Thermo Fisher Scientific Inc., Waltham, MA, USA), 500 ng/ml R-Spondin 1 (Peprotech, Cat. No. 120-38), 100 ng/ml EGF (R&D Systems, Cat. No. 236-EG-01M), and 100 ng/ml Noggin (R&D Systems, Cat. No. 6057-NG-01M), and then passaged every 2 weeks. For in vitro maturation, hIO medium containing recombinant human interleukin 2 (R&D Systems, Cat. No. 202-IL-010) was used for two passages.

## Isolation and culture of ISC$^{3D-hIO}$

The hIOs were dislodged from the Matrigel dome and pipetted up and down to remove the remaining Matrigel fragments surrounding the hIOs. For at least 5 min in a 37 °C water bath, hIOs were digested in 1 ml of 0.25% trypsin-EDTA (Gibco, Cat. No. 25200-072). At the end of digestion, the cells were gently pipetted up and down to break up any aggregated clumps. Four to five rounds of pipetting were sufficient for cell dissociation. Excessive dissociation results in diminished recovery of ISC$^{3D-hIO}$. ISC$^{3D-hIO}$ basal medium (10 ml) was added and centrifuged at $1250 \times g$ for 5 min. The supernatant was carefully discarded, and the pellet was suspended in ISC$^{3D-hIO}$ transfer medium (supplemented with 1 μM Jagged-1 (Anaspec, Cat. No. ANA-AS-61298, Apaspec, Fremont, CA, USA) and 10 μM Y-27632 (R&D Systems, Cat. No. 1254/10) to ISC$^{3D-hIO}$ growth medium). The cells were seeded onto a 35 mm cell culture dish pre-coated with 1% Matrigel or MMC-treated MEF feeder cells. The ISCs$^{3D-hIO}$ were cultured at 37 °C in a 5% CO$_2$ incubator. The ISC$^{3D-hIO}$ full growth medium (supplemented with 2% B27, 10 nM [Leu15]-Gastrin I (Sigma-aldrich, Cat. No. G9145, Merck, Saint Louis, MO, USA), 100 ng/ml Wnt3A, 100 ng/ml EGF, 100 ng/ml Noggin, 500 ng/ml R-Spondin 1, 500 nM A-83-01 (Tocris, Cat. No. 2939, Tocris, Bristol, UK), 10 μM SB202190 (Sigma-aldrich, Cat. No. S7067), 2.5 uM PGE$_2$ (Sigma-aldrich, Cat. No. P0409), 1 mM N-acetylcysteine (Sigma-aldrich, Cat. No. A9165), and 10 mM nicotinamide (Sigma-aldrich, Cat. No. N0636) to ISC$^{3D-hIO}$ basal medium) was replaced every 2 days. The ISCs$^{3D-hIO}$ were passaged at a 1:2 or 1:3 split ratio every 7 days.

## ISC$^{3D-hIO}$ differentiation into the intestinal epithelium

Prepare the Transwell inserts in a 12-well plate by coating them with 250 μl of 1% Matrigel in cold ISC$^{3D-hIO}$ basal medium (2 mM L-glutamine (Gibco, Cat No. 25030-081), 15 mM HEPES (Gibco, Cat No. 15630-080), and 1% penicillin/streptomycin (Gibco, Cat No. 15140-122) in Advanced DMEM/F-12 medium (Gibco, Cat No. 12634-028)). The plate was gently shaken to ensure the Matrigel solution evenly covered the insert surfaces. Then, the plate was incubated for at least 1 h in an incubator at 37 °C with 5% CO$_2$. Dissociated ISCs$^{3D-hIO}$ ($2.5–3.5 \times 10^5$) were seeded onto the Transwell inserts. The ISCs$^{3D-hIO}$ were incubated at 37 °C and 5% CO$_2$, and the ISC$^{3D-hIO}$ full growth medium was changed every 2 days. At confluence, the medium was removed from the inserts by careful pipetting to create an ALI culture. The medium was changed from the outer well plate to the defined minimal medium for ALI differentiation (supplemented with 100 ng/ml EGF, 500 ng/ml R-Spondin 1, 10 μM SB202190, 2.5 μM PGE$_2$, and 10 mM nicotinamide to ISC$^{3D-hIO}$ basal medium) and incubated in an incubator at 37 °C and 5% CO$_2$. The defined minimal medium was changed every 2 days, and the culture was continued for an additional 6–10 days to induce differentiation. The structural development of 2.5D intestinal epithelium was manually analysed by using Image J software. To assess the structural development of 2.5D intestinal epithelium within each condition, we calculated the percentage of structural development of 2.5D intestinal epithelium per image by manually dividing the pixels in the area considered visually similar to the epithelium.

## EdU incorporation assay

The ISC$^{3D-hIO}$ was plated onto 4-well chamber slides (Nunc, Cat No. 177437, Thermo Fisher Scientific Inc., Waltham, MA, USA). After cell attachment in full medium for 2 days, the ISC$^{3D-hIO}$ was grown for 4 days in each growth factor-depleted medium. The ISC$^{3D-hIO}$ was then grown for 24 h in each medium containing 10 μM EdU (Invitrogen, Cat No. C10640). EdU-incorporated ISC$^{3D-hIO}$ was fixed in 4% paraformaldehyde in Dulbecco's phosphate buffered saline (without Ca$^{2+}$ and Mg$^{2+}$). The EdU-positive cells were labelled with the fluorescent dye picolyl azide probe followed by the manufacturer's instructions. For co-staining, samples were labelled by KI67 primary antibody (1:100, BD Bioscience, Cat No. 556003, BD Bioscience, Becton, NJ, USA), which was diluted in 4% bovine serum albumin (BSA, Bovogen Biologicals, Cat No. BSA100, Victoria, Australia) in PBS and DAPI (4′,6-diamidino-2-phenylindole dihydrochloride, Invitrogen, Cat No. D1306) for labelling nuclei. The KI67+ and BrdU+ cells were counted independently in three ISC colonies.

## LIVE/DEAD fluorescence assay

The viability of the ISC$^{3D-hIO}$ was measured using a calcein-AM/Ethidium homodimer 1 LIVE/DEAD assay kit (Invitrogen, Cat No. L3224). After cell washing with DPBS, the optimal volume of staining solution was added according to the manufacturer's instruction, and cells were imaged using a fluorescent microscope (EVOS FL Auto 2, Thermo Fisher Scientific Inc.). The live and dead cells were manually counted, and the ratio of live cells to the total number of cells was used to calculate the survival rate of the ISC$^{3D-hIO}$.

## Whole genome sequencing

For whole genome sequecing of ISC$^{3D-hIO}$ (passage 8, 27, and 54), 100 ng of genomic DNA was used to construct DNA library with TruSeq Nano DNA (Illumina, USA) following the manufacturer's instruction. Multiple libraries were sequenced on an Illumina NovaSeq 6000 using paired-end 150, 6 G reads. Reads were aligned to the reference genome Trimmomatic was used to remove low quality reads to reduce bias. Map the reads to the reference genome (hg38 from UCSC) of choice Burrows-Wheeler Aligner (BWA)[41]. Properly mapped reads were extracted from BAM files after duplicated reads were removed. ngCGH (version 0.4.4) was used to compare two matched BAM data with a window size of 10 kb for copy number estimate. Then, the copy number altered regions were defined by segmentation of the genome using DNAcopy (version 1.74.1)[42].

## Single-cell RNA-sequencing (scRNA-seq)

For scRNA-seq, three independently grown ISC[3D-hIO] cultures were pooled at equal numbers. The ISCs[3D-hIO] were washed three times with Dulbecco's phosphate buffered saline (without $Ca^{2+}$ and $Mg^{2+}$) and treated with 1 ml of 0.25% trypsin-EDTA for 10 min. After cell dissociation, the cell suspension was filtered through a 40-μm cell strainer (Falcon, Cat. No. 352340, Corning, NY, USA) to remove cell aggregates. Single-cell suspensions were washed and resuspended in 0.04% BSA in PBS. Cell viability was determined by trypan blue staining (Gibco, Cat No. 15250-061) and calculated automatically using a Countess™ II system (Thermo Fisher Scientific Inc.). Library construction was performed using the Chromium Next GEM Single Cell 3′ reagent kit v3.1 (10X Genomics) according to the manufacturer's protocol. Briefly, the cells were diluted into Chromium Next GEM Chip G to yield approximately 20,000 single cells. Following library preparation, the libraries were sequenced in multiplex on a Novaseq 6000 sequencer (Illumina) to produce 82,712 reads and 5214 genes per single cell.

## scRNA-seq data analysis

The 10X Genomics software CellRanger (version 3.1) was used to process the raw sequencing data and create gene expression matrices with default parameters. We used published scRNA-seq data from the human intestine from foetal to adult -generated by refs. 18,19, which contained several clearly defined cell populations of reasonable size, to establish a basic cluster of cell types. In other words, it is a relatively large scRNA-seq dataset with excellent annotation. Among them, only the epithelial data of the small intestine of the foetus (6–10 weeks) was selected for comparison with our data. The scRNA-seq analysis was performed using the Scanpy package v1.8. First, predicted doublets were excluded from the analysis using the Scrublet doublet detection pipeline with a threshold of 0.25–0.3. In addition, cells with fewer than 200 genes and greater than 8000 genes were filtered to remove empty droplets and probable doublets, respectively. To account for differences in sequencing depth across samples, we normalised the library size by first dividing the UMI counts by the total UMI counts in each cell and then multiplying by 10,000. For cell clustering, we used marker genes identified by refs. 18,19. with a resolution of 0.4. The scRNA-seq data from this study was integrated using the same method. To confirm the expression of specific marker genes, a normalised dataset without gene filtering was used after dimensionality reduction. We used Spearman's correlation generated by ref. 18. with the scRNA-seq data from this study to estimate the degree of differentiation.

## Bulk RNA-sequencing

The RNA samples were analysed using an Agilent 2100 Bioanalyzer system (Agilent Technologies, Santa Clara, CA, USA). Only high-quality RNA samples (RNA integrity number ≥7.5) were applied in the subsequent preparation of mRNA samples for sequencing. An Illumina TruSeq RNA Sample Preparation Kit v2 (Illumina, San Diego, CA, USA) was used with approximately 0.5–4 μg of total RNA to generate the libraries according to the manufacturer's specifications. RNA sequencing was conducted with an Illumina HiSeq2500 (Illumina, San Diego, CA, USA) following the standard Illumina RNA-Seq protocol by paired-end sequencing with a read length of 100 base pairs.

## Bioinformatic analysis

The NGSQCToolkit v.2.3.3 was used to evaluate the sequence data, Cutadapt v.1.18 were used to trim the adapter sequence from the sample data with the default settings (minimum length = 50 bp, Phred quality threshold score >20). After preprocessing the raw reads, trimmed RNA-seq reads were alligned to the reference genome (GRCh38) using HISAT2 v.2.0.5 with default parameter settings and applying StringTie v.2.1.0, using the reference annotation file to estimate the expression levels of all genes. The expression levels for each transcript were normalised to calculate the sum of mean fragments per kilobase of transcripts per million (FPKM). If the maximum values of this sum across all samples were below 1, the gene was discarded. For identification of DEGs, differences in FPKM values calculated by Cuffdiff were considered significant when the $p$ value was less than or equal to 0.05 and the absolute fold change value was equal to or greater than 2. Using all the protein-coding genes, multidimensional scaling (MDS) analysis was performed to cluster the samples according to their overall similarity of gene expression patterns to determine whether the gene expression patterns between the phenotypic classes could be clearly distinguished. For MDS analysis, the pairwise distances between the samples were determined using the function "dist" (maximum distance measure) in the R v.4.0.2 statistical programming language and plotted using R. The log2 transformation values were used for this analysis, and rows with zero expression in all samples were eliminated. Furthermore, the "hclust" function of the stats package v.3.6.2 was used to perform hierarchical clustering using the maximum distance, and adult small intestine RNA-seq data were downloaded from the public database under accession E-MTAB-1733.

## Lentiviral infection of ISC[3D-hIO]

eGFP-expressing lentivirus (EF1α-Gene X-IRES2-EGFP-IRES-Puro) was purchased from GeneCopoeia (MD, USA). Approximately $2–4 \times 10^5$ ISCs[3D-hIO] were infected with the ISC[3D-hIO] transfer medium supplemented with 8 μg/ml polybrene by centrifugation ($5000 \times g$ for 90 min) in 12-well plates. After centrifugation, a fresh ISC[3D-hIO] transfer medium was added. After 48 h of lentiviral infection, 1 μg/ml puromycin was added to the ISC[3D-hIO] growth medium. To isolate single-cell pedigree lines, trypsinised single cell suspensions were sparsely seeded onto a 1% Matrigel-coated plate. Pedigree lines derived from single cells were obtained after clonogenic cell expansion using a collagenase IV and dispase mixture.

## Xenotransplantation and colonoscopy

Male NIG mice (NOD/SCID deleted IL2Rg gene, 6–12 weeks old; GHBio, Daejeon, Korea) were used in the experiments. The mice were used as a hot EDTA-induced colonic epithelial injury model, as previously described[7]. For orthotopic xenotransplantation, ISCs[3D-hIO] was grown for 4–6 days before being dissociated with trypsin-EDTA. The ISCs[3D-hIO] ($1–5 \times 10^6$ cells) were suspended in Matrigel/Advanced DMEM/F-12 (1:10). A Hundred microliters of ISCs[3D-hIO] suspension was injected into the injured colonic lumen using a colonoscopic injector (Image 1 Hub HD H3-Z; D-Light C; Rigid HOPKINS telescope; Karl Storz, Tuttlingen, Germany; and optimised injector; Vetcom, Gwacheon, Korea) (Matrigel group, $n = 6$; ISCs[3D-hIO] group, $n = 10$). After transplantation, the anal verge was glued with Vetbond Tissue Adhesive (3 M, MN, USA) for 6–12 h. The mice were weighed daily. Each mouse received a colonoscopic examination to track the engraft site at 0, 3, and 14-day post-transplantation. The mice were euthanised on day 14, and the colons were isolated for analysis[7].

## In vivo fluorescence imaging

The visualised ISCs[3D-hIO] xenografts, ISCs[3D-hIO], were stained with 1,1-dioctadecyl-3,3,3,3-tetramethylindotricarbocyanine iodide (DiR, Invitrogen, Cat No. D12731) and xenotransplanted into the EDTA-induced injured colonic lumen. On day 14, the recipient colon was isolated. The colons were monitored using the in vivo imaging system (IVIS Lumina II, Xenogen Corp., CA, USA) at 780 nm (emission)/750 nm (excitation) and were observed using a fluorescent microscope (EVOS FL Auto 2, Thermo Fisher Scientific Inc.)

## Fluorescent stereomicroscopy

To confirm the ISCs[3D-ISX-eGFP-hIO] xenografts, images were acquired using a stereo microscope (SZX16, Olympus, Japan) with bright field and fluorescence (GFP filter). On day 14, the isolated colon tissues were visualised under the same intensity of light and GFP.

## Histopathological analysis

The recipient colon tissues were isolated on 14-day PT, fixed with 10% formalin (Sigma-Aldrich, MO, USA), and incubated with 15–30% sucrose for cryopreservation. Xenograft colons were embedded in Tissue-Tek O.C.T. (Sakura Finetek, Cat No. 4583, Sakura Finetek, CA, USA) compound and sectioned at 10 μm. The tissues were stained with haematoxylin and eosin for histological analysis. For histopathological analysis, Alcian Blue (Abcam, Cat No. 150662, Abcam, Cambridge, UK)-Periodic Acid Schiff (Sigma-aldrich, Cat No. 1.01646.0001) staining was performed on 10-μm colon sections. AB-PAS-stained sections confirmed mucus-secreting goblet cells and mucin. Sections were observed using a microscope (BX53; Olympus, Tokyo, Japan). The colon crypt depth was measured in H&E-stained images using Image J software (Matrigel group, $n = 675$ crypts; ISCs$^{3D\text{-}hIO}$ group, $n = 705$ crypts).

## Immunofluorescence analysis

The cells were rinsed with cold PBS and fixed with 4% paraformaldehyde (PFA, Sigma-Aldrich, Cat No. HT501128) for 15 min at room temperature. Then, the cells were permeabilized with PBS containing 0.1% Triton X-100 (Sigma-Aldrich, Cat No. X100) and blocked with 4% BSA solution for 1 h at room temperature. Then, the cells were incubated overnight in a humid chamber at 4 °C with specific primary antibodies. After incubation, the cells were washed with PBS containing 0.05% Tween 20 (Sigma-Aldrich, Cat No. 1379) and incubated with the secondary antibody for 1 h in the dark. Nuclei were counterstained with 1 mg/ml of DAPI, and cover slips were mounted using fluorescent mounting medium (Dako, Cat No. S3023, Carpinteria, CA). The fluorescence was examined using a confocal microscope (LSM800, Carl Zeiss, Oberkochen, Germany) and a fluorescence microscope (IX51, Olympus, Japan). The xenograft colon sections were incubated with the E-cadherin (R&D systems, Cat No. AF648) and human-specific cytokeratin (BD Biosciences, Cat No. 349205) antibodies overnight at 4 °C. After washing with PBS containing 0.05% Tween 20, the sections were incubated with fluorescently labelled secondary antibodies for 1 h in the dark. Nuclei were counterstained with DAPI. The stained tissues were examined using an EVOS FL Auto 2 system and a confocal microscope (FV1000, Olympus, Tokyo, Japan). The intensity of hCytokeratin+ and DAPI in the xenograft colon were measured with ImageJ 1.53e software. To quantify ALI 4d, 8d, and 12d epithelium thickness, 10 μm sections were stained with an epithelium-specific antibody (ECAD) antibody and imaged using a 20× objective microscope (Olympus). ECAD+ epithelium thickness was randomly measured in 48 areas across three independent sections for each field of view using ImageJ 1.53e software. Additional information on the antibodies used can be found in Supplementary Table 2.

## Quantification of epithelium thickness

The epithelium thickness was randomly quantified straight-line of apical-to-basal measurement in ECAD stained cross-section images by using Image J software. The thickness values were taken from a total of 48 regions in three biological independent sections on days 4, 8, and 12.

## RNA extraction

Cell culture media was removed from culture dishes or plates, and cells were 2–3 times washed with Dulbecco's phosphate-buffered saline (DPBS) containing 0.1% diethyl pyrocarbonate (DEPC). After removing the washing buffer, cells were incubated for 5–10 min with trypsin-EDTA at 37 °C and 5% CO$_2$. After incubation, cells were harvested by centrifugation at $1250 \times g$ for 5 min at RT. RNA was extracted from the cell pellet using a RNeasy kit (Qiagen, Cat No. 74106, Qiagen, Hilden, Germany) followed by the manufacturer's instructions, and then subsequent RNAs obtained were stored at −80 °C.

## Quantitative reverse transcription-polymerase chain reaction (qPCR)

cDNA was synthesised by a Superscript IV cDNA synthesis kit (Invitrogen, Cat No. 18090-050) according to the manufacturer's instruction. A quantitative polymerase chain reaction (qPCR) was performed using the 7500 Fast Real-Time PCR system (Applied Biosystems, Foster City, CA, USA). The relative expression was calculated based on the ΔΔCt method. RNA extracted from the adult human small intestine (hSI) (Clonetech, Cat No. 636539, Takara, Fremont, CA, USA) was used as a positive control. Primers used in this study are listed in Supplementary Table 3.

## Measurement of transepithelial electrical resistance (TEER)

TEER measurements were carried out using an epithelial tissue volt/ohmmeter (EVOM2, WPI, Sarasota, FL, USA) according to the manufacturer's instructions.

## SARS-CoV-2 and cell lines

African green monkey kidney epithelial Vero E6 cells were purchased from the American Type Culture Collection (ATCC CRL-1586; Manassas, VA, USA), and the SARS-CoV-2, KCDC03 (isolated from Korean COVID-19 patient in 2020 and belonging to the A lineage of early Chinese strains), was kindly provided by the National Culture Collection for Pathogens in Korea. The Vero E6 cells were grown in Dulbecco's modified Eagle's medium (DMEM, Gibco) supplemented with 5% FBS (Gibco) and 1% antibiotic-antimycotic (Gibco, Cat No. 15240-062) at 37 °C with 5% CO$_2$. The SARS-CoV-2 KCDC03 strain was propagated in Vero E6 cells in the presence of 1 μg/ml tosyl phenylalanyl chloromethyl ketone (TPCK) trypsin (Sigma-Aldrich, Cat No. 4370185).

## Virus culture

The SARS-CoV-2 was propagated in Vero cells in DMEM without FBS with 1% antibiotic-antimycotic and TPCK trypsin (final concentrations of 1 μg/ml) for 72 h at 37 °C with 5% CO$_2$. Propagated viruses were stored in a freezer at −80 °C for future use. Infectious virus titre were determined using a 50% tissue culture infective dose (TCID$_{50}$) in confluent cells in 96-well microplates. All SARS-CoV-2 experiments were carried out in the KCDC-approved Biosafety Level 3 (BL-3) facility of the Korea Research Institute of Bioscience and Biotechnology (KRIBB) in accordance with institutional biosafety requirements (KRIBB-IBC-20200215).

## Virus inoculation and RNA extraction

The SARS-CoV-2 strain was inoculated onto intestinal epithelial cells ($2.5–3.5 \times 10^5$ cells/well) for 1 h with occasional rocking at a multiplicity of infection (MOI) of 0.01, 0.001. The medium in the transwell inserts was removed and replaced with fresh minimal medium for ALI differentiation after the incubating plates at 37 °C with 5% CO$_2$. The minimal medium was removed 72 h after infection, and total cellular RNA was harvested using the QIAmp Viral RNA Mini kit (Qiagen, Cat No. 52904) following the manufacturer's instructions. Briefly, intestinal epithelial cells from each sample were mixed with 560 μl of buffer AVL containing carrier RNA and incubated for 10 min at room temperature. After the addition of 560 μl of 100% ethanol, carefully apply the mixed solution to the purification columns. And, the solution passed through purification columns by centrifugation at $6000 \times g$ (8000 rpm). The columns were washed sequentially with 500 μl of buffer AW1 and 500 μl of buffer AW2, and RNA was eluted using 50 μl of RNAse-free water. To test the anti-viral effect of selinexor and suramin, 500 nM selinexor or 1 mM suramin were treated for 24 h before viral infection. The following day, the virus was inoculated onto intestinal epithelial cells for 1 h with occasional rocking at a MOI of 0.001.

## Statistics and reproducibility

GraphPad Prism 9.4.1 and Microsoft Excel 2019 were used for data visualisation and analysis. A two-tailed Student's *t*-test was used to

determine the statistical significance of the data, and the results are presented as the mean ± standard deviation (SD) or mean ± standard error (SEM). The number of samples and independent biological experimental repeats were indicated in the figures or figure legends. Differences between means of the crypt depth from individual groups were determined using Welch's *t*-test. The significance is depicted as the *P* value. The difference of endoscopic scores between individual sample groups was determined using a two-tailed non-parametric Mann–Whitney *U* test. The survival analysis was performed by the Kaplan–Meier analysis.

### Reporting summary

Further information on research design is available in the Nature Portfolio Reporting Summary linked to this article.

## Data availability

The source data for all figures are provided as a Source data file. Single cell RNA sequencing data and bulk RNA-seq data that support the findings of this study have been deposited in Gene Expression Omnibus (GEO) with the GSE219018 accession number. All the data generated and/or analysed in this study can be found in the published article or the supplementary information files. Any data in this article are available from the corresponding author (Mi-Young Son; myson@k-ribb.re.kr) upon request, and the requests will be fulfilled within 2 weeks. Source data are provided with this paper.

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

## Acknowledgements

This work was supported by a grant from the National Research Foundation of Korea (NRF) funded by the Ministry of Science, ICT and Future Planning (NRF-2018M3A9H3023077, 2021M3A9H3016046), the Korean Fund for Regenerative Medicine (KFRM) grant funded by the Korean government (Ministry of Science and ICT, Ministry of Health & Welfare, 21A0404L1), a grant from the Technology Innovation Programme (No. 20008777) funded by the Ministry of Trade, Industry & Energy (MOTIE, Korea), a grant (22213MFDS386) from Ministry of Food and Drug Safety in 2023, the KRIBB Research Initiative Programme (KGM4722432), and the Bio & Medical Technology Development Programme of the National Research Foundation (NRF) funded by the Korean government (MSIT) (NBS7942211). The funders had no role in the study design, data collection and analysis, decision to publish, or manuscript preparation.

## Author contributions

Mi-Young Son and Ohman Kwon conceived the project. Mi-Young Son, Ohman Kwon, Hyung-Jun Kwon, and Mi-Ok Lee designed protocols and experiments. Ohman Kwon performed the experiments with the help of Hana Lee, Ye Seul Son, Sojeong Jeon, Won Dong Yu, Naeun Son, Kwang Bo Jung, Eunho Choi, and In-Chul Lee; Ohman Kwon performed the computational analysis with the help of Jaeeun Jung, Hyun-Soo Cho, Chuna Kim, and Dae Soo Kim; Mi-Young Son supervised this study. Mi-Young Son and Ohman Kwon wrote the manuscript. All authors have edited the manuscript.

## Competing interests

The authors declare no competing interests.
