## [Peer Review File · Nature Communications]

REVIEWERS' COMMENTS:

Reviewer #1 (Remarks to the Author)

In this method paper the authors write that they attempted to establish a homogenous and stably expandable intestinal stem cell (ISC) culture system and subsequently differentiate it into 2.5D intestinal epithelium for the development of a highly reproducible and applicable high-throughput screening in vitro intestinal model. The culture system was generated from human pluripotent stem cell (hPSC)-derived 3D human intestinal organoids (hIOs). Generation of 3D hIOs from hPSC was done as previously described in a paper in *Neuropathol Appl Neurobiol* (2017): "Distinctive genomic signature of neural and intestinal organoids from familial Parkinson's disease patient-derived induced pluripotent stem cells". In the present work, 3D hIOs were dissociated into single cells and small cell clumps that were first seeded on a mitomycin C (MMC)-treated feeder. The authors then showed that ISC3D-hIO formed small colonies after attaching to feeder cells, and the colony size rapidly increased due to extensive cell proliferation. After testing several extracellular matrix and media compositions, they confirmed finding by others showing that the cells from dissociated 3D hIOs also proliferate on surface coated with 1% Matrigel and ISC supporting components like Wnt and RSPO1. Cell composition analysis of ISC3D-hIO at single cell resolution showed that the cultures were composed of mainly stem and progenitor cells. The authors differentiated the stem cell enriched cultures into 2.5D intestinal epithelium using minimal medium (without Wnt) and air-liquid interface (ALI) cultures. They then showed examples on how ISC3D-hIO can be used for lentiviral gene editing and transplantation into EDTA-injured colonic epithelium of immunodeficient NOD/SCID deleted IL2Rg gene (NIG) mice. Lastly, the authors infected ALI-differentiated cells from ISC3D-control hIO and ISC3D-mature hIO and showed that cells from ISC3D-mature hIO were more susceptible to SARS-CoV-2 virus infection than the cells from ISC3D-control hIO, due to higher expression of ACE2 in the ALI-differentiated intestinal epithelium from ISC3D-mature hIO.

Model systems like this offer potential valuable tools for applied research, including genetic engineering, regenerative medicine, and disease modelling. Thus, the topic is timely and significant to the field. The main weakness is that, although the authors have performed several relatively advanced techniques, they do not explain or discuss how their approaches differ from other intestinal epithelial organoids system or expansion of undifferentiated, stem cell enriched hIOs. Lack of details in the result section and figure legends make it difficult to follow their intentions, and the proof-of-principles are superficially explored. The work does not show any high-throughput screening (line 217).

Specific comments to results/methods presented in figure 1-6 and Supplementary figures:

Figure 1.

- Are the results from 3 independent experiments with ISC3D-hIO from one donor or 3 different donors?
- Representative images should be supported with quantifications.
- Are the relative expression data in Figure 1e from qPCR? How are cells harvested and prepared for RNA isolation?
- Please provide quantification for Ki67 positive cell detected by Immunofluorescence.
- Line 100: Please rephrase "ISCs 3D hIOs () were completely viable with no cell death. The method used is not sensitive enough to support this conclusion."

Figure 2:

- How many cultures (samples) were used for scRNA seq analysis? The authors write in line 104 that "the dataset comprised 7,034 high-quality cells (Figure 3)". Were the cells diluted into Chromium Next GEM Chip G to yield approximately 5000 single cells for each sample?
- Please describe the adult data (from 20-70 yrs) used in Figure 2b-c. The data set from Elmentaite et al (ref 18) includes fetal and paediatric tissues (up to 14 yrs)?
- Figure 2d is a UMAP plot and not "Dotplot with marker genes used to annotate ISC3D-hIO cell"

types", as stated in the Figure legend. The dotplot is shown in supplementary Figure 5c?

- Figure 2g should be supported by quantifications. Please also provide more details about the experimental design/numbers of replicates analyzed.

Figure 3.

Fig. 3b and supplementary Fig.6. Please provide some quantification and description.

Fig. 3c: Please provide more details about experimental design, hESC, Patient#1, #2 and genome edited cultures. What is the rationale for this comparison, how do you interpret the observations?

Fig3d: Are the genes detected by qPCR or Bulk RNA Sequencing?

Fig3d: Please provide details about numbers of samples in each group.

Fig 3f: Were epithelium thickness at days 4, 8, and 12 after ALI culture analyzed in 48 images?

Fig. 3h. Please describe samples and data generation presented (hPSC (gray, n=4), P0 3D hIO (light blue, n=2), mature 3D hIO (blue, n=3), functional hIECs (pink, n=6), immature ISC3D-hIO (green, n=3), mature ISC3D-hIO (dark green, n=3), immature ALI (yellow, n=3), mature ALI (orange, n = 3), and hSI (red, n=6). Did you perform bulk RNA sequencing of 3 ISC3D-hIO and 3 ALI differentiated cell cultures and compared the data to publicly available dataset of the other samples? How did you harvest the samples for RNA isolation?

Fig 3j. Are the data presented from qPCR analysis or bulk RNA seq? If the latter, a two-tailed t-test is not appropriate.

Figures 4-5.

- Images of the transfected cells expressing GFP in Fig. 4b-e are convincing. Line 170: I suppose genome edited (ISCs3D-hIO-expressing GFP) transplanted into the EDTA-injured colonic epithelium of immunodeficient NOD/SCID deleted IL2Rg gene (NIG)?

- The images in Fig.5f should be supported by quantitative measurement of colon thickness.

- It would be helpful with endoscopic scores to evaluate the healing.

Figure 6.

The result support the conclusion that 2.5D ALI culture from ISC3D-mature hIO was more susceptible to SARS-CoV-2 virus than the intestinal epithelium from ISC3D-control hIO. It would enhance the significance of the work if some mechanistic studies with the virus infected cultures also was provided.

Discussion: Overall, the discussion needs substantial revision. It is fragmented and do not include comparison with other relevant systems and work by others.

Reviewer #2 (Remarks to the Author)

The paper demonstrates a method for 2D monolayer culture of intestinal stem and progenitor cells derived from 3D intestinal organoids. The authors argue that this method of culture is more reproducible and more suitable to scaling for in vitro study of intestinal stem cell biology. Overall, the paper provides a good overview of the potential benefits of maintaining a stem-cell monolayer culture with the potential to differentiate via their '2.5D' culture approach. However, most of components used in their culture have been previously reported and, the ALI approach to cellular differentiation is not especially novel. Here are my comments that might improve the paper. .

General

1. There are several figures where data is presented as a bar-chart, but significance is not indicated within the figure, however, is discussed within the text as if significant. It would be useful to include a marker of non-significance where relevant to make clear those results which did not flag as significant, as on several of these figures SEM seems as if they should be significantly different based on the graph's appearance (examples include figures 3g, S8, and 6e, although there were several other examples).

2. There are several occurrences throughout figures where representative images are provided with no parallel quantification (for example, cell number, culture area, etc).

Specific notes:

Figure 1g) There should also be inclusion of live-dead stain for full media at each of the provided timepoints, for comparison to the -EGF and + PD0325901 conditions

Figure 1c) One condition is labelled as -SB542301 – it is unclear as to what this is. This is potentially an error, as there is later media subtraction of SB202190 mentioned.

Figure 1c) There seems to be an approximately 3 fold increase change in relative occupied area for condition indicated as '-SB' on the bar chart. This has not been noted as significant, which seems surprising based on the appearance of the graph and lack of overlap of SEM of this condition with the mean of the baseline condition. Authors indication that they utilised a two-tailed t-test, which would be expected to mark both positive and negative change from baseline. An explanation of this result in the body of the text may be of value.

Line 65: This data should be included within the main text rather than as a supplementary figure. Additionally, figures S1a, S1b provide qualitative images and cell number as outcomes justifying the selection of Matrigel as the provided matrix. However, both visual outcome and cell number seem largely similar between both collagen and Matrigel conditions based on the cell number outcome – if there is a significant difference between groups, this has not been indicated statistically, and no other assessment of cellular proliferation/survival (such as Ki67 expression, or live-dead staining as used later in the paper) has been included. The justification for use of Matrigel, while well established in other literature, seems to somewhat lack evidence on the basis of this data alone.

Figure 1g: There should also be inclusion of live-dead stain for full media at each of the provided timepoints, for comparison to the -EGF and + PD0325901 conditions

Line 89 / Figure S2c: Authors state "However, all factors affected the survival efficiency of ISC3D-hIO 90 cells after passaging. (Extended Data Fig. S2c, P1)" – For conditions of SB202190 and NAC depletion in particular, the single image provided does not seem to illustrate a noticeable difference from the observed phenotype of the Full M condition – some measure of quantification, such as change in cell number, or evidence of further visual phenotypic deviation following secondary passage, would be of value to support this statement.

Figure 3c: It is unclear what the 'genome edited' condition refers to in this figure. Later in the text, authors state "we used lentiviral gene transduction to create genetically modified ISC3D-hIO lines". If this condition refers to the line produced later in the paper, it would be useful to explain this earlier in the text, as this is left to be inferred by the reader.

Figure 3d: It would potentially be more elegant to either plot absolute expression here, or plot relative expression with reference to hSI, as this figure is aiming to demonstrate a direct comparison to gene expression of in vivo human tissue.

Figure 3h: The ISC 3D control hIO and ISC 3D mature hIO seem to show similar clustering on MDS plot. Would argue that while there were 9 known sample groups included, there are only 8 homogenous sample groups (rather than 9) observable here, with a lack of obvious phenotypic distinction between these two groups compared to others on the plot.

Figure 4 c, d, e: A wild type comparison is included for comparison in figure e, and should ideally also be included in figures c and d to illustrate the phenotypic similarity of the cell line post-editing to its unedited control. Particularly important here as the bright-field of WT versus reporter cell line in figure 4e looks as if it could potentially be phenotypically different based on the representative images chosen. Additionally, a transient fluorescent assay would be useful to visualise the unedited control via fluorescent microscopy, rather than simply providing an image showing the absence of fluorescent signal (which would be expected in a non-fluorescing cell line)

Figure S7d: The normal observed mortality rate for this procedure should also be included as we have no indication as whether either group's mortality significantly deviates from that of normal procedure – is the difference in mortality a product of the EDTA damage or of the transplant procedure? If the argument is being made that ISC transplanted animals see better survival in response to EDTA damage as a product of ISC related intestinal recovery, then a survival curve should really be included here.

Figure 5b: The increase in weight change is only a significant difference at 14 days – the final point out of 14 - why was this timepoint chosen as the final timepoint for the experiment? It would have been informative to see if weight increase equilibrated with the control group over a longer period. If this was a pre-determined time point, this should be indicated in the text. A run of the experiment over a longer time period to see if the recovery, and the observed decrease in mortality, is sustained over a longer period would also be of value.

Figure 5f: a quantification of damaged vs recovered/healthy tissue in control versus transplanted mice would be an important addition here to illustrate the mechanism of the improved recovery (assessed via weight gain) of the experimental groups.

Figure 5g: It would be useful to include a quantification of the number of transplanted cells which

have established within the epithelium. Additionally, some kind of quantification of co-localisation, to verify that regions where the ISC transplant has established show better histopathological recovery from the EDTA damage.

Reviewer #3 (Remarks to the Author)

The authors have created a system for generating enriched cultures of intestinal stem cells from human intestinal organoids in the paper titled 'Chemically defined and scalable culture system for intestinal stem cells derived from human intestinal organoids'. The scheme works out not only to create an enriched iSC population but these cells could be differentiated into mature IE layers with histological and functional villi. The entire scheme has been worked out with chemical medium devoid of feeder layers.

Each step has been demonstrated and validated appropriately. The study is important because of the sheer expandable nature of both ISC and differentiated 2.5 D layers. This system definitely finds uses in disease modeling, microbiome studies, drug screens and as authors have shown genetic remodeling etc.

I was looking for more physiological evidence which is mentioned but lacking in terms of evidence of function during integration of the implants and absorption in the animal studies. Other wise the work appears to be complete in all aspects.

Reviewer #4 (Remarks to the Author)

"Chemically defined and scalable culture system for intestinal stem cells derived from human intestinal organoids"

This is an interesting manuscript that addresses the problem of functionalizing human intestinal organoids for drug development, regenerative medicine, and biological investigations. The dominant model developed in the Clevers lab is based on intestinal organoids in Matrigel which has a low barrier to entry but is problematic with regards to labor-intensive passaging and limited expansion properties. Alternative approaches to propagating stem cells rather than organoids out of the Xian lab circumvents some of these problems but relies on feeder cells and serum-based culture conditions that add a layer of complexity. The present paper describes methods to propagate intestinal stem cells in a feeder-free and defined, serum-free media and their use in regenerative medicine and SARS2 studies. There is much that is interesting with this work, and many glaring questions that must be addressed for this work to contribute to addressing the barriers presented by the dominant intestinal organoid models.

1. Given the authors' choice to focus on iPCS- or ESC-derived intestinal organoids, there needs to be methods describing (not just a reference citation) how they went from iPSCs to definitive endoderm. There also needs to be at least a discussion of why intestinal stem cells can be derived from iPSC/ESCs when this has not been possible for hematopoietic stem cells or epidermal stem cells (vast literature). Part and parcel to this discussion is why the authors did not perform these studies on stem cells derived from intestinal organoids from normal or abnormal gastrointestinal tract? This later system is by far what the industry uses and is seeking alternative and more efficient means of performing. Were these experiments, using the present methods, with normal human intestinal organoids attempted?

2. The authors use several approaches to generate and maintain colonies of ISCs on feeder-free media and conclude that 1% Matrigel is superior to other supports. The data provided in the supplementary figures do not support this superiority, but suggest that other supports work equally well. There are also multiple formulations of "Matrigel" that differ in functional properties, though the authors do not specify which Matrigel they employed. Please clarify these points.

3. Comparisons of the scRNAseq profiles of the ISCs derived from the iPSCs with ref18 suggested similarities with the small intestine from six to eight-week human fetuses. This finding relates to both point #1 and to the regenerative medicine experiments presented in this work. Does this approach from iPSCs yield developmentally "immature" cells akin to cardiomyocytes derived from ESC/iPSCs? Is there a bias from this approach to yield ISCs that have properties of the small intestine? If so, how might this impact attempts at regenerative medicine where functionally distinct portions of the GI tract might be replaced with cells having small intestine properties?

4. Not lastly, but certainly of concern given the history of iPSCs (derivation) and known mutational events impacting the p53 pathway as well as growth factor pathways, there was no effort to characterize the ISCs derived from this protocol (e.g. ISCs vs blood at passage 4 and passage 30) that would reduce this widely held concern.

All of these concerns are ones that may have been addressed off-line of the manuscript, but they are critical to push this technology forward to overcome the blocks presented by status quo intestinal organoids.

RESPONSE TO REVIEWERS' COMMENTS

New experiments/added data:

1. Replacement of ISC^{3D-hIO} bright field images growing on the different types of coating materials (Fig 1c).
2. A graph quantifying cell number (Fig 1d).
3. A result of qPCR analysis (Fig 1e).
4. Replacement of CV staining images and newly added quantification graph (Fig 2a).
5. Replacement of fluorescence images and newly added quantification graphs (Fig 2d).
6. Fluorescence images under Full M at 12, 24, 48 h and quantification graphs (Fig 2e).
7. Graphs quantifying intestinal stem cell marker proteins (Fig 3g).
8. Graphs quantifying differentiated area (Fig 4b).
9. Replacement of labeling (Fig 4c).
10. Replacement of qPCR plot relative expression with reference to hSI (Fig. 4d).
11. Marker of non-significance (ns) in the graph (Fig. 4g).
12. Bright field and fluorescence images of wild-type (WT) ISC^{3D-hIO} (Fig 5c).
13. Bright field and fluorescence images of wild-type (WT) InS^{exp} (Fig 5d).
14. A graph quantifying endoscopic score at 14d PT (Fig 6c).
15. Replacement of H&E staining images and newly added a boxplot quantifying crypt depth (Fig 6f).
16. A graph quantifying fluorescence intensity of hCytokeratin/DAPI (Fig 6g).
17. A result of statistical testing and a marker of non-significance (ns) in the graph (Fig. 7c).
18. A marker of non-significance (ns) in the graph (Fig. 7e).
19. Graphs quantifying the reduction of viral infection by treatment with chemical inhibitors (Fig 7f).
20. Calcein-AM/Etidium homodimer 1 staining images of ISC^{3D-hIO} growing on the different types of coating materials (Fig S1b).
21. KI67 staining images of ISC^{3D-hIO} growing on the different types of coating materials (Fig S1c).
22. Replacing bright field images of ISC^{3D-hIO} growing under the growth factor depleted conditions (Fig S2c).
23. A graph quantifying relative colony sizes of ISC^{3D-hIO} (Fig S2d).
24. Whole-genome profiling of copy number variation (CNV) (Fig S3a, b).
25. Replacing bright field images of 2.5D intestinal epithelium growing under the growth

factor depleted conditions (Fig S6a).

26. A graph quantifying relative fold change of differentiated area (Fig S6b).
27. A Kaplan-Meier plot quantifying survival rate (Fig S7e).
28. A marker of significance in the graph (Fig. S8)

Figure arrangement/layout changes:

1. Original “Fig. 1c-i” have been rearranged to “Fig. 2a-g”, respectively.
2. Original “Fig. 2a-g” have been rearranged to “Fig. 3a-g”, respectively.
3. Original “Fig. 3a-j” have been rearranged to “Fig. 4a-j”, respectively.
4. Original “Fig. 4a-e” have been rearranged to “Fig. 5a-e”, respectively.
5. Original “Fig. 5a-g” have been rearranged to “Fig. 6a-g”, respectively.
6. Original “Fig. 6a-e” have been rearranged to “Fig. 7a-e”, respectively.
7. Original “Supplementary Fig. 1b” have been rearranged to “Fig. 1c, d”, respectively.
8. Original “Supplementary Fig. 2d, e” have been rearranged to “Fig. 2e, f”, respectively.
9. Original “Supplementary Fig. 3a-c” have been rearranged to “Fig. 4a-c”, respectively.
10. Original “Supplementary Fig. 4” have been rearranged to “Fig. 5”, respectively.
11. Original “Supplementary Fig. 5a-d” have been rearranged to “Fig. 6a-d”, respectively.
12. Original “Supplementary Fig. 6a, b” have been rearranged to “Fig. 7a, b”, respectively.
13. Original “Supplementary Fig. 7a-d” have been rearranged to “Fig. 8a-d”, respectively.
14. Original “Supplementary Fig. 8” have been rearranged to “Fig. 9”, respectively.

Reviewers' comments:

Reviewer #1 (Remarks to the Author)

*In this method paper the authors write that they attempted to establish a homogenous and stably expandable intestinal stem cell (ISC) culture system and subsequently differentiate it into 2.5D intestinal epithelium for the development of a highly reproducible and applicable high-throughput screening in vitro intestinal model. The culture system was generated from human pluripotent stem cell (hPSC)-derived 3D human intestinal organoids (hIOs). Generation of 3D hIOs from hPCS was done as previously described in a paper in *Neuropathol Appl Neurobiol* (2017): "Distinctive genomic signature of neural and intestinal organoids from familial Parkinson's disease patient-derived induced pluripotent stem cells". In the present work, 3D hIOs were dissociated into single cells and small cell clumps that were first seeded on a mitomycin C (MMC)-treated feeder. The authors then showed that ISC3D-hIO formed small colonies after attaching to feeder cells, and the colony size rapidly increased due to extensive cell proliferation. After testing several extracellular matrix and media compositions, they confirmed finding by others showing that the cells from dissociated 3D hIOs also proliferate on surface coated with 1% Matrigel and ISC supporting components like Wnt and RSPO1. Cell composition analysis of ISC3D-hIO at single cell resolution showed that the cultures were composed of mainly stem and progenitor cells. The authors differentiated the stem cell enriched cultures into 2.5D intestinal epithelium using minimal medium (without Wnt) and air-liquid interface (ALI) cultures. They then showed examples on how ISC3D-hIO can be used for lentiviral gene editing and transplantation into EDTA-injured colonic epithelium of immunodeficient NOD/SCID deleted IL2Rg gene (NIG) mice. Lastly, the authors infected ALI-differentiated cells from ISC3D-control hIO and ISC3D-mature hIO and showed that cells from ISC3D-mature hIO were more susceptible to SARS-CoV-2 virus infection than the cells from ISC3D-control hIO, due to higher expression of ACE2 in the ALI-differentiated intestinal epithelium from ISC3D-mature hIO.*

Model systems like this offer potential valuable tools for applied research, including genetic engineering, regenerative medicine, and disease modelling. Thus, the topic is timely and significant to the field. The main weakness is that, although the authors have performed several relatively advanced techniques, they do not explain or discuss how their approaches differ from other intestinal epithelial organoids system or expansion of undifferentiated, stem cell enriched hIOs. Lack of details in the result section and figure legends make it difficult to follow their

intentions, and the proof-of principles are superficial explored. The work does not show any high-throughput screening (line 217).

Specific comments to results/methods presented in figure 1-6 and Supplementary figures:

Figure 1.

•Are the results from 3 independent experiments with ISC^{3D-hiO} from one donor or 3 different donors?

Response: We appreciate your valuable comment. The results in Fig. 1 were Fig. 3 independent experiments performed using iPSCs derived from one donor¹. Moreover, it was also confirmed that ISC^{3D-hiO} derived from two different donors (Patient #1-iPSCs and Patient #2-iPSCs)² and hESCs (Fig. S1a) grew in a similar morphologies and patterns (**Q&A Fig. 1**).

Q&A Fig. 2. Representative cell images of ISC^{3D-hiO} (a) and 2.5D intestinal epithelium (b). Black scale bar, 200 μm , White scale bar, 100 μm , and Yellow scale bar, 100 μm .

•Representative images should be supported with quantifications.

Response: As recommended by the Reviewer, we have added quantification analysis data in representative images such as Fig. 1f, Fig. 1g, Fig. 2g, Fig. 3b, Fig. 5c, Fig. 5f, Fig. 5g, Fig. 6f, Fig. S1b, Fig. S2d, Fig. S7e, and Fig. S6b.

•Are the relative expression data in Figure 1e from qPCR? How are cells harvested and prepared for RNA isolation?

Response: We appreciate your valuable comment. In Figure 1e, the relative expression of intestinal stem cell marker gene was confirmed by qPCR analysis. The detailed experimental methods are described below and included in Materials and Methods section.

1. Plating same number of ISC^{3D-hiO} on the 1% Matrigel-coated 6- or 12-well plates in Full M supplemented with 10 μ M Y-27632 and 1 mM Jagged-1.
2. After cell attachment, ISC^{3D-hiO} was grown in Full M or media without each growth factor.
3. The growth medium was changed every other day.
4. At 7–80% confluency of control group (ISC^{3D-hiO} was grown in Full M), 2–3 times washing of ISC^{3D-hiO} with 1X DPBS containing 0.1% DEPC.
5. After complete removal of washing buffer, ISC^{3D-hiO} was incubated 5–10 min with trypsin-EDTA (Thermo Fisher Scientific Inc.) at 37 °C and 5% CO₂.
6. Cells were harvested by centrifugation at 1,250 \times g for 5 min at RT, and then total RNA was extracted from ISC^{3D-hiO} using Qiagen RNeasy kit according to the manufacturer's instructions.

Revised Materials and Methods

RNA extraction

Cell culture media was removed from culture dishes or plates and cells were 2–3 times washed with Dulbecco's phosphate-buffered saline (DPBS) containing 0.1% diethyl pyrocarbonate (DEPC). After complete removal of washing buffer, cells were incubated for 5–10 min with trypsin-EDTA (Thermo Fisher Scientific Inc.) at 37 °C and 5% CO₂. After incubation, cells were harvested by centrifugation at 1,250 \times g for 5 min at RT. RNA was extracted from cell pellet using a RNeasy kit (Qiagen, Hilden, Germany) followed by manufacturer's instructions and then subsequent RNAs obtained were stored at -80 °C.

•Please provide quantification for Ki67 positive cell detected by Immunofluorescence.

Response: As recommended by the Reviewer, we quantified Ki67⁺ and BrdU⁺ cells in ISC^{3D-hiO} colonies of each condition. And, both KI67⁺ and BrdU⁺ cells were significantly reduced by R-spondin 1 or WNT/R-spondin 1 depletion compared to control (Full medium) (**Newly added Fig. 1f**). To clarify this issue, we have added quantification analysis in **Fig. 1(f)** and the revised the text as shown below and the revised portions are indicated with red font in the text.

Newly added Fig. 1f. Quantification of BrdU⁺ and KI67⁺ cells in ISC^{3D-hIO} colonies.

Revised Figure Legend

Fig. 1 (f) immunofluorescence images and quantification analysis in ISC^{3D-hIO} grown in Full M, depletion of WNT3A, RSP01, or WNT3A/RSP01. Yellow scale bar: 50 μ m. **Data** represents the mean \pm SD (n=3). * p < 0.05, ** p < 0.01, *** p < 0.001 using two-tailed t-test.

Revised Materials and Methods

EdU incorporation assay

The ISC^{3D-hIO} was plated onto 4-well chamber slides (Thermo Fisher Scientific Inc.). After cell attachment in full medium for 2 days, the ISC^{3D-hIO} was grown for 4 days in each growth factor-depleted medium. The ISC^{3D-hIO} was then grown for 24 h in each medium containing 10 μ M EdU (Thermo Fisher Scientific Inc.). EdU-incorporated ISC^{3D-hIO} was fixed in 4% paraformaldehyde in Dulbecco's phosphate-buffered saline (without Ca²⁺ and Mg²⁺). The EdU-positive cells were labelled with the fluorescent dye picolyl azide probe, followed by the manufacturer's instructions. For co-staining, samples were labelled by KI67 primary antibody (1:100, BD Bioscience) which was diluted in 4% bovine serum albumin (Bovogen Biologicals, Victoria, Australia) in PBS and DAPI (4',6-diamidino-2-phenylindole dihydrochloride, Thermo Fisher Scientific Inc.) for labelling nuclei. The KI67⁺ and BrdU⁺ cells were counted independently in three ISC colonies.

•*Line 100: Please rephrase "ISCs 3D hIOs () were completely viable with no cell death." The method used is not sensitive enough to support this conclusion.*

Response: Thank you for insightful feedback. Based on your suggestion, we modified the Line 100 sentence as below.

Revised sentence in Results

“and most of the cells were viable and dead cells were hardly found in LIVE/DEAD assay (Extended Data Fig. S3c).”

Figure 2:

•How many cultures (samples) were used for scRNA seq analysis? The authors write in line 104 that “the dataset comprised 7,034 high-quality cells (Figure 3)”. Were the cells diluted into Chromium Next GEM Chip G to yield approximately 5000 single cells for each sample?

Response: Three independently grown ISC^{3D-hIO} cultures were pooled at equal numbers. To library construction, approximately 20,000 single cells of ISC^{3D-hIO} were diluted into the Chromium Next GEM Chip G. Following library preparation, the libraries were sequenced in multiplex on a Novaseq 6000 sequencer (Illumina) to produce 82,712 reads and 5,214 genes per single-cell from 7,034 high-quality cells (Q&A Fig. 2). To clarify this, we have added sentences in the Materials and Methods section. The revised portions are indicated in red font in the manuscript.

Revised Materials and Methods

Single-cell RNA-sequencing

For scRNA-seq, three independently grown ISC^{3D-hIO} cultures were pooled at equal numbers. The ISCs^{3D-hIO} were washed three times with Dulbecco's phosphate buffered saline (without Ca²⁺ and Mg²⁺) and treated with 1 ml of 0.25% trypsin-EDTA for 10 min. After cell dissociation, the cell suspension was filtered through a 40-µm cell strainer (BD Biosciences) to remove cell aggregates. Single-cell suspensions were washed and resuspended in 0.04% BSA in PBS. Cell viability was determined by trypan blue staining (Invitrogen) and calculated automatically using a Countess™ II system (Thermo Fisher Scientific Inc.). Library construction was performed using the Chromium Next GEM Single Cell 3' reagent kit v3.1 (10X Genomics)

according to the manufacturer’s protocol. Briefly, the cells were diluted into Chromium Next GEM Chip G to yield approximately 20,000 single cells. Following library preparation, the libraries were sequenced in multiplex on a Novaseq 6000 sequencer (Illumina) to produce 82,712 reads and 5,214 genes per single-cell from 7,034 high-quality cells.

1_full

Q&A Fig. 2. Summary report produced from 10X genomics scRNA-seq of ISC^{3D-h1O}.

•Please describe the adult data (from 20-70 yrs) used in Figure 2b-c. The data set from Elmentaite et al (ref 18) includes fetal and paediatric tissues (up to 14 yrs)?

Response: Thank you for your thoughtful comments. The public scRNA-seq data used in this study are cells of the human intestinal tract produced by Elmentaite and his colleagues³. Processed scRNA-seq objects are available for download at <https://gutcellatlas.org>. They produced and published scRNA-seq data of foetal and paediatric human intestine tissue in 2020 and adult data in 2021. We used small intestine epithelium scRNA-seq data among the data they produced. Reference 18 in the original manuscript contained only foetal and paediatric

data³, so the paper containing adult data published in Nature (2021) was also added to the references sections⁴.

Revised References

Reference No. 19: Elmentaite, R., Kumasaka, N., Roberts, K. et al. Cells of the human intestinal tract mapped across space and time. Nature 597, 250–255 (2021).

•*Figure 2d is a UMAP plot and not “Dotplot with marker genes used to annotate ISC3D-hIO cell types”, as stated in the Figure legend. The dotplot is shown in supplementary Figure 5c?*

Response: We appreciate the Reviewer for pointing out our mistake. We have changed the legend of Figure 2d as below.

Revised Figure legend in the manuscript

(d) UMAP plot with single cells of ISC3d-hIO colored by cell types.

•*Figure 2g should be supported by quantifications. Please also provide more details about the experimental design/numbers of replicates analyzed.*

Response: We appreciate your valuable comments. From the scRNA-seq data, we confirmed that the majority of ISC^{3D-hIO} were composed of foetal-stage intestinal stem cells and progenitors (**Fig. 2d and e**). Therefore, we examined the expression patterns of intestinal stem cell and progenitor marker genes known from previous studies^{5,6}. In our scRNA-seq data, the majority of ISC^{3D-hIO} were expressed intestinal stem cell and progenitor marker genes (**Fig. 2f**). Next, we tried to do immunofluorescence analysis to examine stem cell and progenitor marker expression in ISC^{3D-hIO}. At the same time, we also verified the differentiated cell markers such as FABP1 as an enterocyte marker, MUC2 as a marker of goblet cells, and CHGA as a marker of enteroendocrine cells. As we expected, fetal ISC marker genes (*LDHB, EIF3E, SOX9*, and

Ki67) were expressed in the majority of ISC^{3D-hIO} (**Fig. 2g**). A small subset of FABP1⁺ enterocytes were detected, but no MUC2⁺ goblet cells and CHGA⁺ enteroendocrine cells were observed at all (**Fig. 2g and S5d**). As the Reviewer suggested, we quantified LDHB⁺, EIF3E⁺, SOX9⁺, Ki67⁺, and FABP1⁺ cells in images of ISC^{3D-hIO} colonies (performed in triplicate), and these results are added in **Fig. 2g**.

Fig. 2g. Quantification of LDHB, EIF3E, SOX9, Ki67 and FABP1 positive cells in ISC^{3D-hIO} colonies (n=3).

Figure 3.

•*Fig. 3b and supplementary Fig.6. Please provide some quantification and description.*

Response: Thank you for your thoughtful comments. We analysed bright field images in biologically replicated data to quantify the degree of 2.5D intestinal epithelium differentiation (n=3). We measured the serpentine structures of 2.5D intestinal epithelium using Image J software, described in the bar graph in modified Fig. 3b and supplementary Fig. 6.

In Fig 3b, a distinctive structure of intestinal epithelium was developed as time went by, and the initial morphogenic processes of 2.5D intestinal epithelium in Minimal M were slightly delayed but not significantly different compared to Full M. However, structural differences between Full M and Minimal M were diminished after the completion of structural development on the 8th day. For quantitative analysis, we used Image J to measure the degree of 2.5D intestinal epithelium development and graphically represented it in Q&A Fig. 3. Quantification data were added in Fig. 3b (right panels).

Q&A Fig. 3. a. The triplicated bright field images of 2.5D intestinal epithelium at days 0, 2, 4, 6, and 8 after ALI culture in full growth medium (Full) or the defined minimal medium (Minimal). **b.** Quantification analysis of 2.5D intestinal epithelium in Full or Minimal. Data represents the mean \pm SD (n=3). * $p < 0.05$, ** $p < 0.01$, *** $p < 0.001$ using two-tailed t-test.

Fig. S6 showed that the development of 2.5D intestinal epithelium was suppressed when certain components were removed from full medium. Specifically, when EGF and R-spondin1 were removed, the development of 2.5D intestinal epithelium was completely diminished. It was also confirmed that impaired development of 2.5D intestinal epithelium was observed when PGE2, SB202190, and nicotinamide were removed (**modified Fig. S6**). For quantitative analysis, we used Image J to measure the degree of 2.5D intestinal epithelium development and graphically represented it in **newly added Fig. S6b**. A detailed description about quantification processes was added to Materials and Methods sections to facilitate understanding.

Fig. S6. The effect of growth factors and cytokines on differentiation via ALI culture. **(a)** The duplicated bright field images of 2.5D intestinal epithelium after 8 days ALI culture in full growth medium or each component depleted medium. **(b)** Quantification analysis of 2.5D intestinal epithelium in full growth medium or each component depleted medium. Data represents the mean \pm SD (n=4). * $p < 0.05$, ** $p < 0.01$, *** $p < 0.001$ using two-tailed t-test.

Revised paragraph in Materials and Methods

ISC^{3D-hIO} differentiation into intestinal epithelium

The structural development of 2.5D intestinal epithelium was manually analysed by using Image J software.

•Fig. 3c: Please provide more details about experimental design, hESC, Patient#1, #2 and genome edited cultures. What is the rationale for this comparison, how do you interpret the observations?

Response: We appreciate your comment. Cell line-to-line differences between donors, genetic background, and experimental variability generate differences in differentiation potential, cell morphology, maturity, and functionality^{7,8}. To determine whether the differentiation of ISC^{3D-hIO} into 2.5D intestinal epithelium by ALI culture method would be generalised for various hPSC lines, we differentiated five different ISC^{3D-hIO} lines into 2.5D intestinal epithelium under

ALI culture condition (one human embryonic stem cell line, H9-hESC (in Fig. 3c); three independent human induced pluripotent stem cell lines, CRL-iPSC (in Fig. 3b), Patient #1-iPSC (in Fig. 3c), Patient #2-iPSC (in Fig. 3c); one genome-edited hiPSC line, EGFP-expressing CRL-iPSC (in Fig. 4e). As we have shown in Fig. 3c, different ISC^{3D-hiO} lines were successfully differentiated into 2.5D intestinal epithelium without any morphological differences. These results mean that ISC^{3D-hiO} differentiation into 2.5D intestinal epithelium under ALI culture conditions is a common process independent of cell line-to-line variation.

•Fig3d: Are the genes detected by qPCR or Bulk RNA Sequencing? Please provide details about numbers of samples in each group.

Response: Thank you for your comment. The genes in Fig. 3d were detected by qPCR analysis. In addition, three biological replicated samples (each sample was technically duplicated for qPCR analysis, n=6) in ALI-differentiated cells at days 0, 4, 8, and 12 after air exposure group were analysed in Fig 3d and RNA extracted from the adult human small intestine (hSI) (Clontech, Fremont, CA, USA) was used as a positive control.

Revised Figure legend in the manuscript

Fig. 3. (d) Relative expression of stem cell marker genes (*LGR5*, *CD44*, *MKI67*, *SOX9*, *ASCL2*, *OLFM4*, *AXIN2*, and *CTNNB*), and differentiated cells (*VILI*, *ECAD*, *FABP1*, *KRT20*, *LCT*, *LYZ*, and *MUC2*) in ALI-differentiated cells at days 0, 4, 8, and 12 after air exposure and hSI. Data represents the mean \pm SEM (*n=6*).

•Fig 3f: Were epithelium thickness at days 4, 8, and 12 after ALI culture analyzed in 48 images?

Response: We appreciate your comments. To quantify the thickness of 2.5D intestinal epithelium, we randomly selected 48 regions from ECAD-stained cross-section images in three biologically independent samples on days 4, 8, and 12. From the randomly selected images, we measured the height of apical-to-basal epithelium using Image J software and described in a bar graph with overlapping dots in Fig. 3f. Every single dot indicated that the height of apical-to-basal epithelium derived from the randomly selected region. A detailed description about quantification processes was added to Materials and Methods sections to facilitate understanding.

Revised paragraph in Materials and Methods

Quantification of epithelium thickness

The epithelium thickness was a randomly quantified straight line of apical-to-basal measurement in ECAD-stained cross-section images using Image J software. The thickness values were taken from 48 regions in three biological independent sections on days 4, 8, and 12.

•Fig. 3h. Please describe samples and data generation presented (hPSC (gray, n=4), P0 3D hIO (light blue, n=2), mature 3D hIO (blue, n=3), functional hIECs (pink, n=6), immature ISC3D-hIO (green, n=3), mature ISC3D-hIO (dark green, n=3), immature ALI (yellow, n=3), mature ALI (orange, n = 3), and hSI (red, n=6). Did you perform bulk RNA sequencing of 3 ISC3D-hiO and 3 ALI differentiated cell cultures and compared the data to publicly available dataset of the other samples? How did you harvest the samples for RNA isolation?

Response: We would like to express our deep gratitude for the Reviewer's comment. In this study, we newly generated bulk RNA sequencing dataset of immature ISC^{3D-hiO} (green, n=3), mature ISC^{3D-hiO} (dark green, n=3), immature ALI (yellow, n=3), mature ALI (orange, n = 3). The remaining bulk RNA sequencing dataset containing hPSC (gray, n=4), P0 3D hIO (light blue, n=2), mature 3D hIO (blue, n=3), functional hIECs (pink, n=6), and hSI (red, n=6) was generated and publicly uploaded in our previous reports^{1,2}. The detailed experimental methods for harvesting the immature and mature ISC3D-hiO and ALI samples are described below and we modified the Materials and Methods section.

1. Plating same number of immature or mature ISC^{3D-hiO} on the 1% Matrigel-coated 6- or 12-well plates or 12-well transwell plate for ALI culture and cells were grown in Full M supplemented with 10 μM Y-27632 and 1mM Jagged-1.

2. After cell attachment, ISC^{3D-hiO} and ALI cells were grown in Full M and medium was changed every other day.

3-1. At 7-80% confluency of immature or mature ISCs^{3D-hiO}, 2-3 times washing of ISCs^{3D-hiO} with 1X DPBS containing 0.1% DEPC

3-2-1. At confluence, the medium was removed from the inserts for ALI culture and minimal medium was added in the outer well plate.

3-2-2. The minimal medium was changed every two days and the culture was continued for an additional 6–10 days to induce differentiation.

3-2-3. After 8 days starting ALI culture, 2-3 times washing of immature or mature ALI with 1X DPBS containing 0.1% DEPC.

4. After complete removal of washing buffer, ISC^{3D-hIO} or ALI was incubated 5-10 min with trypsin-EDTA (Thermo Fisher Scientific Inc.) at 37 °C and 5% CO₂.

5. Cells were harvested by centrifugation at 1,250 xg for 5 min at RT, and then total RNA was extracted from ISC^{3D-hIO} using Qiagen RNeasy kit according to the manufacturer's instructions.

Revised Materials and Methods

RNA extraction

Cell culture media was removed from culture dishes or plates, and cells were 2-3 times washed with Dulbecco's phosphate-buffered saline (DPBS) containing 0.1% diethyl pyrocarbonate (DEPC). After removing the washing buffer, cells were incubated for 5-10 min with trypsin-EDTA (Thermo Fisher Scientific Inc.) at 37 °C and 5% CO₂. After incubation, cells were harvested by centrifugation at 1,250 xg for 5 min at RT. RNA was extracted from the cell pellet using a RNeasy kit (Qiagen, Hilden, Germany) followed by the manufacturer's instructions, and then subsequent RNAs obtained were stored at -80 °C.

•*Fig 3j. Are the data presented from qPCR analysis or bulk RNA seq? If the latter, a two-tailed t-test is not appropriate.*

Response: We appreciate your comment. The genes in Fig. 3d were detected by qPCR analysis to validate bulk RNA-seq data. Therefore, we thought that application of a two-tailed t-test is appropriate to analysis Fig. 3j.

Figures 4-5.

•*Images of the transfected cells expressing GFP in Fig. 4b-e are convincing. Line 170: I suppose genome edited (ISCs3D-hIO-expressing GFP) transplanted into the EDTA-injured colonic epithelium of immunodeficient NOD/SCID deleted IL2Rg gene (NIG)?*

Response: We appreciate your valuable comment. The fluorescence reporter cells, $ISCs^{3D-ISX-eGFP-hIO}$, used for the transplantation study in Fig. 5 were different from the reporter cell-line, GFP-expressing $ISCs^{3D-hIO}$ by infection with GFP-expressing lentivirus described in Fig. 4. $ISCs^{3D-ISX-eGFP-hIO}$ in Fig. 5 were generated from the enhanced green fluorescence protein-expressing 3D hIOs (hIOs- ISX^{eGFP}), which were differentiated from ISX^{eGFP} reporter hPSC expressing eGFP under the intestine lineage-specific promoter (intestine-specific homeobox; ISX) published in our previous report⁹. When comparing three types of cell lines, including wild type (WT) $ISCs^{3D-hIO}$, $ISCs^{3D-ISX-eGFP-hIO}$ in Fig. 5, and EGFP-expressing $ISCs^{3D-hIO}$ in Fig. 4, there were no differences observed in terms of cell morphology and other characteristics (Q&A Fig. 4).

Q&A Fig. 4. Morphologies of ISC^{3D-hIO} , $ISC^{3D-ISX-eGFP-hIO}$, and EGFP-expressing $ISCs^{3D-hIO}$

•The images in Fig.5f should be supported by quantitative measurement of colon thickness.

Response: We appreciate your valuable comment. As per the Reviewer's comment, we quantitatively assessed the colon crypt depth of the xenograft tissues (Q&A Fig. 5). To clarify this issue, we have added quantification analysis in Fig. 5(f) and revised the text as shown below. The revised portions are indicated with red font in the text.

Q&A Fig. 5. Crypt depth analysis of mouse tissues

(Matrigel group: n=675 from three mice, ISC^{3D-hIO} group: n=705 from eight mice)

Revised text in Figure legends

Fig. 5. (f) The box and scatterplots of crypt depths of Matrigel (n=675 crypts from three mice) and ISC^{3D-hIO} (n=705 crypts from eight mice) transplanted mouse tissues (H&E staining). Each dot represents a crypt; each column depicts an individual sample. Bar represents the mean of each sample. *** $p < 0.001$, Welch's t-test.

Revised paragraph in Materials and Methods

Histopathological analysis

The colon crypt depth was measured in H&E stained images using Image J software.

Quantification and statistical analysis

All *in vitro* experiments were repeated at least three times, and the results are presented as the mean \pm standard error (SEM). A two-tailed Student's t-test was used to determine the statistical significance of the data. Differences between means of the crypt depth from individual groups were determined using Welch's t-test. The significance is depicted as the *P* value. The difference in endoscopic scores between individual sample groups was determined using a two-tailed non-parametric Mann-Whitney U test.

•*It would be helpful with endoscopic scores to evaluate the healing.*

Response: We greatly appreciate your suggestions. We established criteria for endoscopic analysis reflecting intestinal epithelial damage and epithelial recovery after transplantation according to clinical endoscopic evaluation¹⁰. Based on this analysis criterion, we measured the recovery evaluation score from the colonoscopy result of the transplanted mice (**Q&A Fig. 6** and **Q&A videos 1 and 2**). To clarify this issue, we have newly added quantification analysis in **Fig. 5(c)** and the revised text as shown below. The revised portions are indicated with red font in the text.

Endoscopic score				
EDTA injury	None	Mild	Moderate	Severe
Erythema	0	1	2	3
Bleeding	0	1	2	3
Intestinal epithelial damage (erosion)	0	1	2	3
Epithelial damage area	0	1	2	3
Total score				

Endoscopic score				
Transplantation	None	Mild	Moderate	Severe
Erythema and bleeding	0	1	2	3
Intestinal epithelial damage (erosion)	0	1	2	3
Inflammation from wound	0	1	2	3
Total score				

Transplantation	Endoscopic score					
Matrigel	Erythema and bleeding	erosion	inflammation from wound	total		average
1						4.333333333
2	1	1	0	2		
3	1	2	3	6		
4						
5	0	2	3	5		
6						
Transplantation	Endoscopic score					
ISC	Erythema and bleeding	erosion	inflammation from wound	total		average
1						1.25
2	0	2	3	5		
3						
4	0	0	0	0		
5	0	0	0	0		
6	0	1	0	1		
7	0	0	0	0		
8	1	1	0	2		
9	0	1	1	2		
10	0	0	0	0		

Q&A Fig. 6 (newly added **Fig. 5c**). Endoscopic score analysis in the transplanted mice. $*p < 0.05$, A two-tailed non-parametric Mann-Whitney U test.

Revised text in Figure legends

Fig. 5c. Endoscopic score of the transplanted mice. Matrigel group (n=3), ISC^{3D-HIO} group (n=8).

Revised paragraph in Materials and Methods

Quantification and statistical analysis

The difference of endoscopic scores between individual sample groups was determined using a two-tailed non-parametric Mann-Whitney U test.

Figure 6.

•The result support the conclusion that 2.5D ALI culture from ISC3D-mature hIO was more susceptible to SARS-CoV-2 virus than the intestinal epithelium from ISC3D-control hIO. It would enhance the significance of the work if some mechanistic studies with the virus infected cultures also was provided.

Response: We appreciate your insightful comments. As shown in our manuscript, the amount of SARS-CoV2 infection was correlated to the expression level of its receptor, ACE2 (Fig. 6e and Fig. S8), which is also well-known for a receptor of SARS-CoV-2¹¹. However, the expression levels of cell surface proteins such as the serine proteases ADAM17 and TMPRSS2, which facilitate the SARS-Cov-2 viral infection, were not correlated to the amount of SARS-CoV2 infection level (Fig. S8). It means that the expression level of ACE2 is a deterministic factor for SARS-CoV2 infection in the 2.5D intestinal epithelium model system. To validate the importance of ACE2 for SARS-CoV2 infection, we treated chemical inhibitors that can block the interaction between ACE2 and SARS-CoV-2 or perturb nuclear export of ACE2 peptides (Q&A Fig. 7).

Q&A Fig 7. Pathophysiological mechanism of SARS-CoV-2 infection via interaction with ACE2.

The ISCs^{3D-hIO} were pre-treated with either selinexor or suramin for 24 hrs before SARS-CoV-2 infection. As we expected, either selinexor, a novel selective inhibitor of nuclear export,

reduces SARS-CoV-2 infection, or suramin, which binds and inhibits infection of SARS-CoV-2 through both spike protein-heparan sulfate and ACE2 receptor interactions, treatment suppressed SARS-CoV-2 infection in 2.5D intestinal epithelium (**Newly added Fig. 6f**).

Fig. 6f. Perturbation effects by selinexor or suramin on the interaction between SARS-CoV-2 and ACE2 in 2.5D intestinal epithelium. Data represents the mean \pm SEM ($n=8$). *** $p < 0.001$ using a two-tailed t -test.

Further experiments are needed to reveal the underlying molecular mechanism of SARS-CoV-2 infection in host cells. Still, current results suggest that the infection of SARS-CoV-2 virus is mediated through the interaction with the ACE2 receptors in the 2.5D intestinal epithelium. We have included these results in Fig. 6f. and revised the Materials and Method, Results, and Discussion sections. The revised portions are indicated in red font in the manuscript. We appreciate your insightful advice and will continue to update our progress in future studies.

Revised paragraph in Result

(line -) On the contrary, the infection of the virus was significantly inhibited when treated with chemical drugs that can disrupt the interaction between ACE2 and SARS-CoV-2 (Fig. 6e).

Revised paragraph in Discussion

Consistently, we also confirmed the essential role of ACE2 as a viral receptor in ALI-

differentiated intestinal epithelium from ISC^{3D-hIO} because the reduction of viral infection was observed when treating with chemical inhibitors that inhibit plasma membrane expression of ACE2 or chemical inhibitors that interfere with the binding between SARS-CoV-2 and ACE2 (ref추가). Therefore, the sensitivity to SARS-CoV-2 infection in human intestinal epithelium is determined by the expression level of the ACE2, and it can be verified that SARS-CoV-2 is less likely to infect children(1, 2) who have less developed SARS-CoV-2 binding sites of ACE2(3, 4).

Revised paragraph in Materials and Method

To test the anti-viral effect of selinexor and suramin, 500 nM selinexor or 1 mM suramin were treated for 24 h before viral infection. The following day, the virus was inoculated onto intestinal epithelial cells for 1 h with occasional rocking at MOI of 0.001.

Discussion: Overall, the discussion needs substantial revision. It is fragmented and do not include comparison with other relevant systems and work by others.

Response: We appreciate the Reviewer's comment. We have revised the "Discussion" section to incorporate the Reviewer's recommendations; the revised portions are indicated in red in the text.

Revised paragraph in Discussion

In this study, we attempted to establish a homogenous and stably expandable ISC culture system and subsequently differentiate it into 2.5D intestinal epithelium for the development of a highly reproducible and applicable high-throughput screening *in vitro* intestinal model. The novel 2D ISC^{3D-hIO} culture system supports an enriched population of ISCs and early progenitors under fully defined culture media and feeder-free conditions, rapid propagation, long-term maintenance with simple passaging, efficient cryopreservation of ISC^{3D-hIO} with multiple cycles of freezing and thawing, and highly reproducible differentiation into functional cells when needed. Thus, the ISC^{3D-hIO} is a desirable cell source for applications such as *in vitro* model systems for mimicking intestinal physiology, disease modelling, genome editing, and regenerative medicine via cell transplantation.

To mimic *in vivo* intestinal physiology, these stem cells must be differentiated into intestinal epithelial cells, such as absorptive and secretory cells. Although 3D hIOs are a well-known model system containing differentiated cells, an intestinal epithelial model system capable of quantitative assessment is required to overcome the limitations of 3D hIOs, they cannot be used for conventional intestinal assays using 2D monolayer cultures(5). Our method of ALI culture to differentiate ISC^{3D-hIO} into 2.5D intestinal epithelium provides a controllable *in vitro* intestinal model with easy access to the lumen, reduced batch variation, and compatibility with functional assays. Evidently, it was observed that a villus-like structure grew from a flat monolayer over time, and the expression levels of differentiated cell marker genes and barrier function also increased during morphogenesis. **Furthermore, the 2.5D intestinal epithelium model system enables easy manipulation of intestinal stem cells without needing stem cell sorting using cell sorting technologies such as FACS. These advantages allow it to be easily integrated with bio-engineering technologies such as micropatterned plates and gut-on-a-chip, not just transwell plates.**

Based on these advantages, SARS-CoV-2 infection was confirmed, and viral RNAs were expressed in ALI-differentiated intestinal epithelium from ISC^{3D-hIO}, a model host cell system for SARS-CoV-2 infection. Because ACE2, the receptor for SARS-CoV-2, is highly expressed in ALI-differentiated intestinal epithelium from ISC^{3D-mature hIO} compared with that in ALI-differentiated intestinal epithelium from ISC^{3D-control hIO}, SARS-CoV-2 infection occurred more frequently in ALI-differentiated intestinal epithelium from ISC^{3D-mature hIO}. ACE2 plays a key role as a viral receptor for SARS-CoV-2, and TMPRSS2-mediated proteolytic cleavage of the ACE2 cytoplasmic tail increases endosomal internalisation(6, 7). **Consistently, we also confirmed the essential role of ACE2 as a viral receptor in ALI-differentiated intestinal epithelium from ISC^{3D-hIO} because the reduction of viral infection was observed when treating with chemical inhibitors that inhibit plasma membrane expression of ACE2 or chemical inhibitors that interfere with the binding between SARS-CoV-2 and ACE2 (ref 추가).** Therefore, the sensitivity to SARS-CoV-2 infection in human intestinal epithelium is determined by the expression level of the ACE2, and it can be verified that SARS-CoV-2 is less likely to infect children(1, 2) who have less developed SARS-CoV-2 binding sites of ACE2(3, 4). Our data demonstrate that ISC^{3D-hIO} preserves maturity-specific gene expression patterns reflecting their hIO origin, such as fetal-like and adult-like maturation characteristics. In line with these findings, the adjustable differentiation of ISC^{3D-hIO} derived from hIOs

representing a particular state in an intestinal epithelial monolayer can be used for various applications, including intestinal morphogenesis and disease modelling.

Furthermore, when the global gene expression pattern was analysed using bulk RNA-seq, *in vivo* intestinal metabolism and nutrient transport-related terms were highly enriched in the ALI-differentiated intestinal epithelium than in the ISC^{3D-hIO}. However, there are limitations regarding setting and use as a more physiologically mimetic intestine model because the differentiated cells in the 2.5D intestinal epithelium from ISC^{3D-hIO} have yet to reach maturity. **Indeed, we confirmed that the maturity and functionality of the 2.5D intestinal epithelium from ISC^{3D-hIO} were lower than those of the 2D functional intestinal epithelium directly differentiated from hPSCs, as recently reported by our group (ref). These phenomena were presumed to be related to the abundance of growth factors or cytokines, which maintain stemness; now we are conducting further studies to assess structural development of 2.5D intestinal epithelium to find the best conditions to enhance the maturity of the 2.5D intestinal epithelium. Furthermore, it is necessary to develop a co-culture system using various stromal cells that can closely mimic the native environment of the human intestine in future studies(8, 9).**

In this study, we demonstrated the possibility of using ISC^{3D-hIO} as an obvious target for genetic engineering and as a transplant source for intestinal diseases. Despite the need for additional functional characterisation and application studies, we suggest that ISC^{3D-hIO} is easy to handle and a suitable cell source for various applications, including genetic engineering. We generated eGFP reporter ISC^{3D-hIO} cell lines that can efficiently differentiate into hIOs and intestinal epithelium while retaining their morphology. hIOs have mostly been used in previous studies for *in vivo* transplantation for damaged tissue regeneration(10-12). These studies suggest that LGR5⁺ stem cells play a key role in cell engraftment and tissue regeneration and can be used for stem cell therapy in regenerative medicine. However, isolating LGR5⁺ stem cells from 3D hIO is technically difficult; most cases have transplanted whole organoids or mechanically decomposed organoid fragments(10, 11, 13, 14). ISC^{3D-hIO} is primarily composed of ISCs and early progenitors and can be transplanted without needing a separate process. Transplanted ISC^{3D-hIO} was rapidly engrafted into the damaged region and successfully reconstituted the intestinal epithelium within 2 weeks. Furthermore, xenotransplant recipient mice recovered their body weight faster than Matrigel-transplanted control mice, implying a beneficial effect of ISC^{3D-hIO} transplantation. Although more research into transplantation

methods and long-term monitoring is needed, the current study suggests that ISC^{3D-hIO} transplantation is a promising treatment method for patients with gastrointestinal epithelial disorders.

In conclusion, we established an ISC^{3D-hIO} culture system that could be highly enriched for rapidly expanding ISCs and early progenitors derived from 3D hIOs. Our study suggests the feasibility of the ISC^{3D-hIO} culture system for applied research, including genetic engineering, regenerative medicine, and disease modelling. It is also conceivable that, with further refinement, the differentiation method into 2.5D intestinal epithelium using ALI Transwell will be compatible with studies such as drug absorption, **drug toxicity**, and microbe-epithelium interaction analysis.

Reviewer #2 (Remarks to the Author)

The paper demonstrates a method for 2D monolayer culture of intestinal stem and progenitor cells derived from 3D intestinal organoids. The authors argue that this method of culture is more reproducible and more suitable to scaling for in vitro study of intestinal stem cell biology. Overall, the paper provides a good overview of the potential benefits of maintaining a stem-cell monolayer culture with the potential to differentiate via their '2.5D' culture approach. However, most of components used in their culture have been previously reported and, the ALI approach to cellular differentiation is not especially novel. Here are my comments that might improve the paper.

General

1. There are several figures where data is presented as a bar-chart, but significance is not indicated within the figure, however, is discussed within the text as if significant. It would be useful to include a marker of non-significance where relevant to make clear those results which did not flag as significant, as on several of these figures SEM seems as if they should be significantly different based on the graph's appearance (examples include figures 3g, S8, and 6e, although there were several other examples).

Response: We appreciate your nice comment. According to your suggestion, we conducted a statistical test again and added a marker to indicate statistical significance. Moreover, a marker of non-significance (ns) is also included on the graph to make clear statistical results (**Q&A Fig 8**).

Q&A Fig. 8. Add a marker of significance or non-significance in Fig 3g, S8, and 6e.

2. There are several occurrences throughout figures where representative images are provided with no parallel quantification (for example, cell number, culture area, etc).

Specific notes:

Figure 1g) There should also be inclusion of live-dead stain for full media at each of the provided timepoints, for comparison to the -EGF and + PD0325901 conditions

Response: We appreciate your valuable comment. As you suggested, we included LIVE/DEAD staining results for full media at each time point for comparison to the -EGF and +PD0325901 conditions. And we quantified the percentage of survival cell rate from images of the LIVE/DEAD assay, and these results are newly added in **Fig. 1g**.

Revised Fig. 1g in the manuscript

Fig. 1g. Live (Calcein-AM)/Dead (EthD-1) analysis of ISC^{3D-hiO} in Full medium or after depletion of EGF, or treatment with PD0325901 at 0, 12, 24, and 48 h, and representative images of ISC^{3D-hiO} grown in Full M, depletion of EGF, treatment with 10 nM PD0325901, or treatment with 100 nM PD0325901. White scale bar: 100 μ m. (left panels) Histograms of the survival rate of ISC^{3D-hiO} at 0, 12, 24, and 48 h.

Figure 1c) One condition is labelled as -SB542301 – it is unclear as to what this is. This is potentially an error, as there is later media subtraction of SB202190 mentioned.

Response: Thank you for your thoughtful comment. ‘SB542301’ was a typo, so we corrected it to ‘SB202190’.

Revised Fig. 1c in the manuscript

Figure 1c) There seems to be an approximately 3 fold increase change in relative occupied area for condition indicated as '-SB' on the bar chart. This has not been noted as significant, which seems surprising based on the appearance of the graph and lack of overlap of SEM of this condition with the mean of the baseline condition. Authors indication that they utilised a two-tailed t-test, which would be expected to mark both positive and negative change from baseline. An explanation of this result in the body of the text may be of value.

Response: Thank you for your thoughtful comment. As you point out, there was an approximately 3-fold increase change in -the SB group, but the variation between the individual samples was too large to show any statistical significance. We believed that these results come from batch-by-batch variation, so we attempted to prepare the additional sample and analysed staining data again (Q&A Fig. 9). As a result of re-analysis, we observed a reduction in batch-by-batch variation from the bright field images (Q&A Fig. 9a), and the CV staining also shown the consistent results (Q&A Fig. 9b, **Modified Fig 1c**). Furthermore, we verified again that there was no statistically significant difference between the Full medium condition and SB202190 depleted condition (Q&A Fig. 9b, **Modified Fig 1c**).

Q&A Fig 9. Representative images of bright field (a Fig S2c merged with red box images) and crystal violet (CV) stained ISC^{3D-h1O} colonies (b, **Modified Fig 1c**) with depletion of a single component from growth media, and quantification of the occupied area by ISC^{3D-h1O}. Data represent the mean \pm SEM ($n=3$).

Line 65: *This data should be included within the main text rather than as a supplementary figure. Additionally, figures S1a, S1b provide qualitative images and cell number as outcomes justifying the selection of Matrigel as the provided matrix. However, both visual outcome and cell number seem largely similar between both collagen and Matrigel conditions based on the cell number outcome – if there is a significant difference between groups, this has not been indicated statistically, and no other assessment of cellular proliferation/survival (such as Ki67 expression, or live-dead staining as used later in the paper) has been included. The justification for use of Matrigel, while well established in other literature, seems to somewhat lack evidence on the basis of this data alone.*

Response: We appreciate your insightful comments on our research. To find out the most suitable matrix for ISC^{3D-hiO} culture, we conducted multiple analysis of cell proliferation and survival, including cell counting, Ki67 expression, live-dead staining, and qPCR analysis. First, we observed stable ISC^{3D-hiO} growth when cultured on culture dishes coated with Collagen type I or Matrigel (Fig. S1b). Consistently, we also found that higher growth rate of ISC^{3D-hiO} grown on the Collagen type I- or Matrigel-coated culture dishes 1 week after seeding an equal number of ISC^{3D-hiO} (Fig. S1c). Furthermore, ISC^{3D-hiO} exhibited enhanced cell growth on the culture dishes coated with 10 µg/ml Collagen type I or 1% Matrigel compared to the culture dishes coated with 50 µg/ml Collagen type I or 5% Matrigel (Fig S1c). However, little difference was observed between 10 µg/ml Collagen type I or 1% Matrigel coating conditions in terms of ISC^{3D-hiO} growth. Also, we were able to confirm a 100% cell survival rate without detecting any dead cells and a stable expression of proliferation marker protein such as KI67, when culturing ISC^{3D-hiO} on the 10 µg/ml Collagen type I- or 1% Matrigel-coated culture dishes (Fig. S1d, e). These data indicated that ISC^{3D-hiO} maintenance on the culture dishes coated with 10 µg/ml Collagen type I or 1% Matrigel is the best condition for ISC^{3D-hiO} culture. However, the ISC^{3D-hiO} grown on the 1% Matrigel-coated culture dishes exhibited significantly higher expression of stemness marker genes (*CD44*, *SOX9*, and *MKI67*) compared to the ISC^{3D-hiO} grown on the 10 µg/ml Collagen type I -coated culture dishes (Fig S1f). Based on qPCR analysis, we primarily used 1% Matrigel rather than 10 µg/ml Collagen type I as a coating material for culturing ISC^{3D-hiO}. We have updated Fig S1 as shown below.

Fig S1. Establishment of ISC^{3D-hiO} culture system. **a.** Morphologies of ISC^{3D-hiO} generated from hESC-derived hiOs on the feeder or 1% Matrigel-coated plate at day 7. Black scale bar: 200 µm. Yellow scale bar: 50 µm. **b.** Efficiency of cell attachment at P0, P1, and P2, Black scale bar: 200 µm. **c.** Average cell number 1 week after cell seeding; Data represents the mean ± SEM (n = 6). **d.** Live (Calcein-AM)/Dead (EthD-1) analysis of ISC^{3D-hiO}, Orange scale bar: 200 µm. **e.** Immunofluorescence images of KI67, White scale bar: 100 µm. **f.** Relative expression of stem cell marker genes (*LGR5*, *CD44*, *SOX9*, *MKI67*, *AXIN2*, and *CTNNB*), and differentiated cells (*VIL1*, *ECAD*, *FABP1*, *KRT20*, *LYZ*, and *MUC2*). Data represents the mean ± SEM (n = 4). **p* < 0.05, ***p* < 0.01, ****p* < 0.001 using a two-tailed *t*-test.

Figure 1g: There should also be inclusion of live-dead stain for full media at each of the provided timepoints, for comparison to the -EGF and + PD0325901 conditions

Response: We appreciate to your valuable comment. As you suggested, we included LIVE/DEAD staining results for full media at each time point for comparison to the -EGF and +PD0325901 conditions. And we quantified the percentage of survival cell rate from images of the LIVE/DEAD assay, and these results are newly added in **Fig. 1g**.

Revised **Fig. 1g** in the manuscript

Fig. 2g. Live (Calcein-AM)/Dead (EthD-1) analysis of ISC^{3D-hIO} in Full medium or after depletion of EGF, or treatment with PD0325901 at 0, 12, 24, and 48 h, and representative images of ISC^{3D-hIO} grown in Full M, depletion of EGF, treatment with 10 nM PD0325901, or treatment with 100 nM PD0325901. White scale bar: 100 μ m. (left panels) Histograms of the survival rate of ISC^{3D-hIO} at 0, 12, 24, and 48 h.

Line 89 / Figure S2c: *Authors state, “However, all factors affected the survival efficiency of ISC3D-hIO 90 cells after passaging. (Extended Data Fig. S2c, P1)” – For conditions of SB202190 and NAC depletion in particular, the single image provided does not seem to illustrate a noticeable difference from the observed phenotype of the Full M condition – some measure of quantification, such as change in cell number, or evidence of further visual phenotypic deviation following secondary passage, would be of value to support this statement.*

Response: We appreciate to your insightful comments. As you indicated, that single image of ISC^{3D-hIO} was under the Full M or growth factor depleted conditions. To provide more robust evidence to support our observations, we repeatedly tested the effect of growth factor depletion on the ISC^{3D-hIO} growth and measured the size of colonies in each condition for quantitative analysis (n=3). As we described in the original manuscript, most components did not show any difference in cell growth before passaging (P0 in Newly added Fig. S2c). However, most factors affected the survival or growth of ISC^{3D-hIO} cells after passaging (P1 in Newly added Fig. S2c).

Newly added Fig. S2. c. Representative images of ISC^{3D-h10} colonies with depletion of a single component from Full M at P0 and P1. Black scale bar: 200 μ m. White scale bar: 100 μ m. **d.** Relative ISC^{3D-h10} colony size to Full M. Data represents the mean \pm SEM ($n = 3$). * $p < 0.05$, ** $p < 0.01$, *** $p < 0.001$ using a two-tailed t -test.

Especially in SB202190 or NAC depleted conditions, statistically significant differences in colony size were not overserved before passing P0 in Newly added Fig. S2c). As the serial passaging continues, we are able to observe a gradual decline in cell culture efficiency with the reduction of average colony size, and the cell growth rate becomes slower compared to Full M (Q&A Fig. 10). From these results, we concluded that every component in Full M is required to stable long-term maintenance of ISC^{3D-h10}. Thank you for your invaluable contribution.

Q&A Fig. 10. Representative images of ISC^{3D-hIO} colonies with depletion of SB202190 or NAC from Full M at P0, P1, and P2. Black scale bar: 200 μ m.

Figure 3c: *It is unclear what the ‘genome edited’ condition refers to in this figure. Later in the text, authors state “we used lentiviral gene transduction to create genetically modified ISC3D-hIO lines”. If this condition refers to the line produced later in the paper, it would be useful to explain this earlier in the text, as this is left to be inferred by the reader.*

Response: We appreciate your nice comment. To clarify the information on genome edited cell-line, we replaced “genome edited cell-line” with a “EGFP-expressing stable hiPSC line.” The EGFP-expressing stable reporter hiPSC lines (ISX^{eGFP}-hiPSCs) were generated and described in our previous report⁹.

Revised Results

(line 124-128) Only five factors (RSPO1, EGF, PGE2, SB202190, and nicotinamide) were present in the designated minimal medium for ALI differentiation when ISC3D-hIO were differentiated into 2.5D intestinal epithelium **from human embryonic stem cells (hESCs), patient derived human induced pluripotent stem cells (Patient #1-, and Patient #2-hiPSCs) or EGFP-expressing stable hiPSC line** (Fig. 3b, c, and Extended Data Fig. S6).

Figure 3d: *It would potentially be more elegant to either plot absolute expression here, or plot relative expression with reference to hSI, as this figure is aiming to demonstrate a direct comparison to gene expression of in vivo human tissue.*

Response: We appreciate your helpful comment. According to your suggestion, we plot the relative expression level of intestinal marker genes with reference to hSI. And we updated Fig. 3d with the newly analysed graph.

Fig. 3d. Relative expression of stem cell marker genes (*LGR5*, *CD44*, *MKI67*, *SOX9*, *ASCL2*, *OLFM4*, *AXIN2*, and *CTNNB*) and differentiated cells (*VIL1*, *ECAD*, *FABP1*, *KRT20*, *LCT*, *LYZ*, and *MUC2*) in ALI-differentiated cells at days 0, 4, 8, and 12 after air exposure and hSI. Data represent the mean \pm SEM (n=6).

Figure 3h: The ISC 3D control hIO and ISC 3D mature hIO seem to show similar clustering on MDS plot Would argue that while there were 9 known sample groups included, there are only 8 homogenous sample groups (rather than 9) observable here, with a lack of obvious phenotypic distinction between these two groups compared to others on the plot.

Response: We appreciate your helpful comment. In our understanding, it seems like the immature ISC^{3D-hIO} (green, n=3) and mature ISC^{3D-hIO} (dark green, n=3) are clustered too closely together on the MDS plot, making it appear as 8 homogenous sample groups (rather than 9). Upon your suggestion, it is possible to put the immature ISC^{3D-hIO} and mature ISC^{3D-hIO} together as one group and to represent 8 groups in total. However, due to the distinct cellular origins of immature ISC^{3D-hIO} and mature ISC^{3D-hIO}, as well as the presence of numerous differentially expressed genes (Q&A Fig. 11), they were divided into separated groups on the MDS plot despite appearing close to each other. Based on these reasons, we were represented as 9 distinct sample groups in the manuscript.

Q&A Fig. 11. Pairwise correlation matrix of immature ISC^{3D-hIO} and mature ISC^{3D-hIO}.

Figure 4 c, d, e: A wild type comparison is included for comparison in figure e, and should ideally also be included in figures c and d to illustrate the phenotypic similarity of the cell line post-editing to its unedited control. Particularly important here as the bright-field of WT versus reporter cell line in figure 4e looks as if it could potentially be phenotypically different based on the representative images chosen. Additionally, a transient fluorescent assay would be useful to visualise the unedited control via fluorescent microscopy, rather than simply providing an image showing the absence of fluorescent signal (which would be expected in a non-fluorescing cell line)

Response: We appreciate reviewer’s valuable comments. To illustrate the phenotypic similarity of the cell line post-editing to its unedited control cell line, we added the bright-field images of wild type ISC^{3D-hIO} and wild type InS^{exp} (Fig. 4c and d). The overall characteristics, such as cell morphology, growth rate, and degree of differentiation, weren’t significantly different between wild-type ISC^{3D-hIO} and genome edited ISC^{3D-hIO} (Fig 4c-e). We included these data in Fig. 4c-e as follows. We appreciate your input.

Fig 4. Generation of an eGFP-expressing ISC^{3D-hIO} reporter cell line. **b.** Representative images of ISC^{3D-hIO} after infection, selection & expansion, low density cell seeding, and expansion to form colonies. White scale bar: 100 μ m. **c-d.** Cell growth images of individual colonies (**c**) and 3D expandable intestinal spheres (**d**) are grown from wild-type or single EGFP-expressing cells. White scale bar: 100 μ m. Yellow scale bar: 50 μ m. **e.** 2.5D intestinal epithelium via ALI differentiation. White scale bar: 100 μ m.

In addition, following the Reviewer's suggestion, we transiently expressed the EGFP proteins in ISC^{3D-hIO}. Although only a small number of cells expressing EGFP fluorescence by the low transfection efficiency, we never found any morphological difference between wild type ISC^{3D-hIO} and transient EGFP-expressing ISC^{3D-hIO} (Q&A Fig. 12). Based on these results, we concluded that there is no phenotypic difference between wild type ISC^{3D-hIO} lines and EGFP-expressing ISC^{3D-hIO} lines.

Q&A Fig 12. Morphologies of WT ISC^{3D-hIO}, WT InS^{exp}, transient EGFP-expressing ISCs^{3D-hIO}, and transient EGFP-expressing InS^{exp}.

Figure S7d: *The normal observed mortality rate for this procedure should also be included as we have no indication as whether either group's mortality significantly deviates from that of normal procedure – is the difference in mortality a product of the EDTA damage or of the transplant procedure? If the argument is being made that ISC transplanted animals see better survival in response to EDTA damage as a product of ISC related intestinal recovery, then a survival curve should really be included here.*

Response: We appreciate Reviewer's insightful comments. To confirm the difference in the mortality rate for transplantation procedure, we measured endoscopic scores for EDTA-induced intestinal epithelium injury models to evaluate individual-to-individual variation before ISC^{3D-hIO} transplantation, and there were no significant differences among Matrigel only and ISC^{3D-hIO} transplantation groups (**Q&A Fig. 13a**). It means that the transplantation procedures were normally working without any variation between individuals by EDTA damage and as well as the mortality rate of mice at day 0 was 9.26%, indicating successful

disease modelling with EDTA treatment (Q&A Fig. 13b). Therefore, the increase of survival rate after ISC^{3D-hIO} transplantation is expected to come from the regenerative effect on intestinal epithelium. ISC^{3D-hIO} transplantation for cell therapy has to employ ECM (Matrigel) to improve stem cell survival and engraftment rate efficacy. In accordance with the recommendations of the IACUC to minimise animal sacrifice, a no-treated group (without Matrigel and ISC^{3D-hIO}) was not prepared. Therefore, we set the control group as Matrigel-only implantation, and the ISC^{3D-hIO} group had ISC^{3D-hIO} + Matrigel transplanted. To evaluate the intestinal recovery by ISC^{3D-hIO} transplantation, the Matrigel only or ISC^{3D-hIO} + Matrigel were transplanted into the EDTA-damaged mice. The survival rate in the ISC^{3D-hIO} + Matrigel transplanted mice was significantly higher than that in the Matrigel transplanted mice (Q&A Fig. 13c). To clarify this issue, we have added survival rate analysis in Supplementary Fig. 7(e) and the revised the text as shown below and the revised portions are indicated with red font in the text.

Q&A Fig. 13. Regenerative effects of ISC^{3D-hIO} transplantation on EDTA-damaged intestinal

epithelium. **a.** Endoscopic score measurement at day 0. (Matrigel: Score 8.17, n = 6; ISC^{3D-hIO}: Score 8.3, n = 10) **b.** The success rate of EDTA-induced intestinal epithelium injury model at day 0. (n=54) (c) Kaplan-Meier plot for survival rate analysis. (Matrigel, n = 6; ISC^{3D-hIO}, n = 10).

Revised text in Figure legends

Supplementary **Fig. 7e.** Kaplan-Meier plot for survival rate analysis. (Matrigel, n = 6; ISC^{3D-hIO}, n = 10)

Revised paragraph in Materials and Methods

Quantification and statistical analysis

We applied time-to-event analysis to determine the survival of each mouse population. Survival plots were estimated by the Kaplan–Meier method.

Figure 5b: *The increase in weight change is only a significant difference at 14 days – the final point out of 14 - why was this timepoint chosen as the final timepoint for the experiment? It would have been informative to see if weight increase equilibrated with the control group over a longer period. If this was a pre-determined time point, this should be indicated in the text. A run of the experiment over a longer time period to see if the recovery, and the observed decrease in mortality, is sustained over a longer period would also be of value.*

Response: We sincerely appreciate the Reviewer's valuable questions. We referred to the 2018 Cell Stem Cell (Sugimoto et al.) about generating an EDTA-induced intestinal epithelial injury model. They showed intestinal epithelial damage was almost reduced within 2 weeks in the colon organoids transplanted group, in which colon organoids were derived from human colon tissues (**Q&A Fig. 14**). Also, we observed rectal bleeding and anal obstruction due to wounds as a disease phenotype daily in the pre-exam. At an average of 10d post-transplantation (PT), all mice don't have any disease phenotypes or weight loss. Based on these results, we set 2 weeks as the endpoint. We added key references to the Materials and Methods section.

Reconstruction of the Human Colon Epithelium *In Vivo*

Authors

Shinya Sugimoto, Yuki Ohta, Masayuki Fujii, ..., Tetsuya Nakamura, Takanori Kanai, Toshiro Sato

Graphical Abstract

Q&A Fig. 14. Sugimoto et al., Cell Stem Cell. 2018

In our results, the histological degree (H&E staining) of epithelial recovery in the Matrigel-transplanted group on 14d PT was similar to that of healthy mice. However, some inflammation and redness caused by the wound remained in the endoscopic analysis. We also showed that mucin secretion was less expressed in the Matrigel-transplanted group than ISC^{3D-hIO} group (newly updated **Fig. 5f**). Moreover, in this experiment, the Matrigel transplant group experienced rapid weight gain (**Q&A Fig. 15**, red dotted box) and gloss hair from the 10th day because of the diet increase, and there were no signs of death after that. We found that mortality in the intestinal epithelial injury model was weight-dependent, with no concern for mortality from when weight gain occurred. As the Reviewer's comment, we plan to perform a valuable experiment to observe long-period after cell transplantation in follow-up research focusing on regenerative therapy.

Q&A Fig. 15. Our results of EDTA-induced injury models

Figure 5f: a quantification of damaged vs recovered/healthy tissue in control versus transplanted mice would be an important addition here to illustrate the mechanism of the improved recovery (assessed via weight gain) of the experimental groups.

Response: As stated by the Reviewer, tissue quantification of control (Matrigel) and transplanted (ISC^{3D-hiO}) mice was necessary to elucidate the mechanism of weight gain in the experimental group. Epithelial recovery by ISC^{3D-hiO} transplantation can be demonstrated through quantitative assessment of crypt depth at 14 days PT (Q&A Fig. 14)¹⁰. Sugimoto et al. reported that the human organoids formed larger crypt structures than the surrounding recipient mouse crypts in the xenograft model. The Q&A Fig. 16 associated with the newly updated Fig. 5f shows a quantitative measure of crypt depth in our study. The epithelium was lost after the EDTA injury, as shown in the intestinal histology on day 0. After 14 days, the epithelium of the Matrigel group is restored, like the healthy mice. In contrast, the ISC^{3D-hiO} -transplanted group shows higher crypt depth reflecting the human crypt depth. Alcian blue (AB) and periodic acid-Schiff (PAS), detect acidic and neutral mucin, respectively. The AB-PAS staining pattern differs between mice and humans^{12, 13}, the mouse colonic goblet cells are AB+ at the crypt bottom and

PAS+ at the upper crypt, whereas human colonic goblet cells are homogeneously stained by both AB and PAS (**Q&A Fig. 16**). Therefore, we also confirmed that mucin secretion was expressed from the bottom to the upper part of the crypt in ISC^{3D-hiO} group using AB-PAS staining. We observed death with weight loss when the epithelium fails to recover after mechanical wounds. Therefore, the weight gain post-transplantation supports the results of epithelial repair (**Fig. 5b**), and the results about the sizes of crypts were included in the newly updated **Fig. 5f**. Thank you for your invaluable contribution.

Q&A Fig. 16. Crypt depth analysis of mouse tissues in our study.

(Matrigel group: n=675 from three mice, ISC^{3D-hiO} group: n=705 from eight mice) **Q&A Fig.**

Newly updated Fig. 5f. Histological analysis of the xenograft tissues (H&E staining, upper) and histopathology of the xenograft colon (AB-PAS, bottom). Black scale bar, 200 µm. **The**

box and scatterplots of crypt depths of Matrigel (n=675 crypts from three mice) and ISC^{3D-hIO} (n=705 crypts from eight mice) transplanted mouse tissues (H&E staining). Each dot represents on crypt; each column depicts an individual sample. Bar represents the mean of each individual sample.

Figure 5g: *It would be useful to include a quantification of the number of transplanted cells which have established within the epithelium. Additionally, some kind of quantification of co-localisation, to verify that regions where the ISC transplant has established show better histopathological recovery from the EDTA damage.*

Response: We appreciate Reviewer's valuable comments. To confirm the distribution of transplanted cells within the damaged colonic epithelium, the fluorescence intensities of human cytokeratin (hCytokeratin) were measured in Matrigel or ISC^{3D-hIO} transplantation group. In addition, we also quantitatively evaluated the colonic epithelium recovery by ISC^{3D-hIO} transplantation through quantification of human-specific cytokeratin to nuclei (DAPI) (**newly added Fig. 5h**). We have newly added this information in the Results and Figures. The revised portions are indicated in red font in the manuscript.

Revised text in Results

The fluorescence intensities of hCytokeratin were significantly increased in ISC^{3D-hIO} transplantation group compared to those in Matrigel group (**Fig. 5h**).

Newly added Fig. 5h. Fluorescence intensity of hCytokeratin/DAPI (n of fields = 10 in Matrigel group, n of fields = 12 in ISC^{3D-hIO} group)

Reviewer #3 (Remarks to the Author)

The authors have created a system for generating enriched cultures of intestinal stem cells from human intestinal organoids in the paper titled 'Chemically defined and scalable culture system for intestinal stem cells derived from human intestinal organoids'. The scheme works out not only to create an enriched inSC population but these cells could be differentiated into mature IE layers with histological and functional villi. the entire scheme has been worked out with chemical medium devoid of feeder layers.

Each step has been demonstrated and validated appropriately. The study is important because of the sheer expandable nature of both ISC and differentiated 2.5 D layers. This system definitely finds uses in disease modeling, microbiome studies, drug screens and as authors have shown genetic remodeling etc.

I was looking for more physiological evidence which is mentioned but lacking in terms of evidence of function during integration of the implants and absorption in the animal studies. Other wise the work appears to be complete in all aspects.

Response: We appreciate Reviewer's valuable comment. As the Reviewer mentioned, measuring the functionalities of the transplanted cells on the damaged intestinal tissue is an important factor for therapeutic effect evaluation. However, measuring the functionality of transplanted cells in a living mouse is a challenging task. This is because the transplanted site is localised, finding the location of transplanted cells is difficult, and the available assay methods for real-time tracking of intestinal epithelial cell function are limited. Despite these limitations, we can find some evidence of intestinal function recovery from our experimental data. Firstly, we can observe rapid weight gain in the ISC^{3D-hIO} transplanted mice. Since weight gain is generally strongly associated with intestinal function, the rapid increase in mice weight implies a functional recovery of the damaged intestine (**Fig 5b**). Secondly, we observed the presence of goblet cells scattered within the transplanted intestinal tissue through AB-PAS staining. Goblet cells are terminally differentiated cells that constitute part of the intestinal epithelium. The terminally differentiated functional goblet cells are rarely observed in the immature and foetal intestine in humans, rats, and chick¹⁴. However, we observed a significant presence of goblet cells within the ISC^{3D-hIO} transplanted intestinal tissue based on the AB-PAS staining (**Fig 5f**). This indicates a high-level differentiation potential of the ISC^{3D-hIO} when transplanted in damaged intestinal tissue and suggests that it is performing normal functions as

part of the intestinal epithelium. We appreciate your insightful advice and will continue to update our progress in future *in vivo* studies aimed at directly measuring the functionality of the ISC^{3D-h10} in transplanted animals.

Reviewer #4 (Remarks to the Author)

"Chemically defined and scalable culture system for intestinal stem cells derived from human intestinal organoids."

This is an interesting manuscript that addresses the problem of functionalizing human intestinal organoids for drug development, regenerative medicine, and biological investigations. The dominant model developed in the Clevers lab is based on intestinal organoids in Matrigel which has a low barrier to entry but is problematic with regards to labor-intensive passaging and limited expansion properties. Alternative approaches to propagating stem cells rather than organoids out of the Xian lab circumvents some of these problems but relies on feeder cells and serum-based culture conditions that add a layer of complexity. The present paper describes methods to propagate intestinal stem cells in a feeder-free and defined, serum-free media and their use in regenerative medicine and SARS2 studies. There is much that is interesting with this work, and many glaring questions that must be addressed for this work to contribute to addressing the barriers presented by the dominant intestinal organoid models.

1. Given the authors' choice to focus on iPSC- or ESC-derived intestinal organoids, there needs to be methods describing (not just a reference citation) how they went from iPSCs to definitive endoderm. There also needs to be at least a discussion of why intestinal stem cells can be derived from iPSC/ESCs when this has not been possible for hematopoietic stem cells or epidermal stem cells (vast literature). Part and parcel to this discussion is why the authors did not perform these studies on stem cells derived from intestinal organoids from normal or abnormal gastrointestinal tract? This later system is by far what the industry uses and is seeking alternative and more efficient means of performing. Were these experiments, using the present methods, with normal human intestinal organoids attempted?

Response: We appreciate your insightful comment. During the development of vertebrate embryos and differentiation of human pluripotent stem cells (hPSCs), TGF- β /nodal/activin signaling is well-known for essential signaling to induce definitive endoderm (DE) specification¹⁵⁻¹⁷. Various methods to induce differentiation from hPSCs to DE have been developed, but co-treatment of Activin A, a nodal-related TGF- β molecule, with increasing concentrations of defined foetal bovine serum (dFBS) was most efficiently and robustly

working to induce DE specification as previously described^{1, 16, 18}. Therefore, we adopted that method to induce DE differentiation, and hPSCs were treated with 100 ng/ml Activin A for 3 days in RPMI1640 medium with increasing concentrations of 0, 0.2%, and 2% dFBS. And then, DE cells were further differentiated into hindgut (HG) cells by incubation in DMEM/F12 supplemented with 2% dFBS, 500 ng/ml FGF4, and 500ng/ml WNT3A (or 3 μ M CHIR99021) for 4 days. Through these differentiation processes, hPSCs, including hESCs and hiPSCs, were differentiated into the endodermal lineage, while haematopoietic stem cells derived from the mesodermal lineage or epidermal stem cells derived from the ectodermal lineage were not generated. Furthermore, it was reported that the efficiency of DE or HE differentiation from hPSCs was very high, with over 85% differentiation efficiency¹⁸.

As Reviewer's comment, we also tried to develop cultivation methods for intestinal stem cells derived from the normal or abnormal gastrointestinal tract. Although not included in this manuscript, isolation, and cultivation of ISCs^{3D-hCO} from human normal tissue (adult stem cells)-derived colon organoids (hCOs) have developed (**Q&A Fig. 17**). Adult stem cell-derived 3D colonoids (hCOs) grown in Matrigel were dissociated and plated onto the feeder cells. As shown in **Q&A Fig. 17**, the attached intestinal stem cells yield colonies comprised of highly proliferative cells and consistently propagated (*Upper panels in Q&A Fig. 17*), and ISCs^{3D-hCO} were passaged in conditions of exponential growth in culture condition (*Bottom panels in Q&A Fig. 17*).

From these results, we anticipate that the ability to maintain intestinal stem cells derived from normal or patient tissue will enable the academic and industrial applications such as disease modelling, drug screening, and pre-clinical trials. To do that, subsequent research is being conducted to establish a method for culturing ISCs^{3D-hCO} isolated from normal- and cancer-derived colonoids in a feeder-free condition with the chemically defined medium. We appreciate your insightful advice and will continue to update our progress in future studies.

Q&A Fig. 17. Intestinal stem cells (ISC^{3D-hCO}) derived from human 3D colon organoids (colonoids)

2. The authors use several approaches to generate and maintain colonies of ISCs on feeder-free media and conclude that 1% Matrigel is superior to other supports. The data provided in the supplementary figures do not support this superiority, but suggest that other supports work equally well. There are also multiple formulations of "Matrigel" that differ in functional properties, though the authors do not specify which Matrigel they employed. Please clarify these points.

Response: We appreciate your insightful comments on our research. To find out the most suitable matrix for ISC^{3D-hIO} culture, we conducted multiple analysis of cell proliferation and survival, including cell counting, Ki67 expression, live-dead staining, and qPCR analysis. First, we observed stable ISC^{3D-hIO} growth when cultured on culture dishes coated with Collagen type I or Matrigel (**Fig. S1b**). Consistently, we also found that higher growth rate of ISC^{3D-hIO} grown on the Collagen type I- or Matrigel-coated culture dishes 1 week after seeding an equal number of ISC^{3D-hIO} (**Fig. S1c**). Furthermore, ISC^{3D-hIO} exhibited enhanced cell growth on the culture dishes coated with 10 µg/ml Collagen type I or 1% Matrigel compared to the culture dishes coated with 50 µg/ml Collagen type I or 5% Matrigel (**Fig S1c**). However, little difference was

observed between 10 $\mu\text{g/ml}$ Collagen type I or 1% Matrigel coating conditions in terms of $\text{ISC}^{3\text{D-hIO}}$ growth. Also, we were able to confirm a 100% cell survival rate without detecting any dead cells and a stable expression of proliferation marker protein such as KI67, when culturing $\text{ISC}^{3\text{D-hIO}}$ on the 10 $\mu\text{g/ml}$ Collagen type I- or 1% Matrigel-coated culture dishes (**Fig S1d, e**). These data indicated that $\text{ISC}^{3\text{D-hIO}}$ maintenance on the culture dishes coated with 10 $\mu\text{g/ml}$ Collagen type I or 1% Matrigel is the best condition for $\text{ISC}^{3\text{D-hIO}}$ culture. However, the $\text{ISC}^{3\text{D-hIO}}$ grown on the 1% Matrigel-coated culture dishes exhibited significantly higher expression of stemness marker genes (*CD44*, *SOX9*, and *MKI67*) compared to the $\text{ISC}^{3\text{D-hIO}}$ grown on the 10 $\mu\text{g/ml}$ Collagen type I -coated culture dishes (**Fig. S1f**). Based on qPCR analysis, we primarily used 1% Matrigel rather than 10 $\mu\text{g/ml}$ Collagen type I as a coating material for culturing $\text{ISC}^{3\text{D-hIO}}$. We have updated Fig S1 as shown below.

Fig S1. Establishment of $\text{ISC}^{3\text{D-hIO}}$ culture system. a. Morphologies of $\text{ISC}^{3\text{D-hIO}}$ generated from hESC-derived hIOs on the feeder or 1% Matrigel-coated plate at day 7. Black scale bar: 200 μm . Yellow scale bar: 50 μm . **b.** Efficiency of cell attachment at P0, P1, and P2, Black scale bar: 200 μm . **c.** Average cell number 1 week after cell seeding; Data represents the mean

± SEM (n = 6). **d.** Live (Calcein-AM)/Dead (EthD-1) analysis of ISC^{3D-hIO}, Orange scale bar: 200 µm. **e.** Immunofluorescence images of KI67, White scale bar: 100 µm. **f.** Relative expression of stem cell marker genes (*LGR5*, *CD44*, *SOX9*, *MKI67*, *AXIN2*, and *CTNNB*), and differentiated cells (*VIL1*, *ECAD*, *FABP1*, *KRT20*, *LYZ*, and *MUC2*). Data represents the mean ± SEM (n = 4). **p* < 0.05, ***p* < 0.01, ****p* < 0.001 using a two-tailed *t*-test.

3. Comparisons of the scRNAseq profiles of the ISCs derived from the iPSCs with ref18 suggested similarities with the small intestine from six to eight-week human fetuses. This finding relates to both point #1 and to the regenerative medicine experiments presented in this work. Does this approach from iPSCs yield developmentally "immature" cells akin to cardiomyocytes derived from ESC/iPSCs? Is there a bias from this approach to yield ISCs that have properties of the small intestine? If so, how might this impact attempts at regenerative medicine where functionally distinct portions of the GI tract might be replaced with cells having small intestine properties?

Response: We appreciate your insightful comment on our research. As you mentioned, hPSC-derived cells and organoids, such as cardiomyocytes and intestinal organoids derived from hESCs/iPSCs, are more similar to immature or foetal tissues undergoing development^{19, 20}. Therefore, the ISC^{3D-hIO} retained immature characteristics since it originated from hPSC-derived 3D hIOs. This immaturity of the ISC^{3D-hIO} exhibited similar gene expression patterns to the small intestine of 6-8 weeks-old fetuses based on the scRNA-seq analysis (**Fig. 2b, c**). However, every single cell of the ISC^{3D-hIO} fully differentiated into intestinal epithelial cells (**Fig. 2a**, EPCAM⁺ and CDH1⁺ cells), and lineage specific differentiation into the small intestine had already occurred during the small intestinal organoid formation step²¹. Especially during the scRNA-seq analysis, the gene expression profile of all individual cells was analysed rather than filtering out some cells expressing marker genes related to the small intestine. These indicated that the majority of ISC^{3D-hIO} exhibited characteristics of the small intestinal epithelial cells due to the high purity of the ISC^{3D-hIO}, not come from analytical bias.

Finally, we agree with your comment that there may be limitations on the transplantable sites due to the ISC^{3D-hIO} having small intestine properties. Fortunately, we can differentiate hPSCs into various parts of the GI tract, such as stomach, duodenum, jejunum, ileum, and colon, and also culture region specific organoids derived from various tissue resident stem cells²². For instance, 2D colon stem cells isolated from 3D colon organoids from a fresh biopsy of normal tissue can be successfully cultured in our system (**Q&A Fig. 17**), making it feasible to utilise them as a cell therapy for colon disease. Therefore, it is anticipated that region specific stem

cells can be transplanted into damaged regions, and the therapeutic effects are expected to be higher than mismatched transplantation. In this study, the priority was to conceptually describe the regenerative effects through the transplantation of the ISC^{3D-hIO}. Hence, we didn't use the site matched organoids for the transplantation.

All of these concerns may have been addressed off-line in the manuscript, but they are critical to pushing this technology forward to overcome the blocks presented by status quo intestinal organoids.

4. Not lastly, but certainly of concern given the history of iPSCs (derivation) and known mutational events impacting the p53 pathway as well as growth factor pathways, there was no effort to characterize the ISCs derived from this protocol (e.g. ISCs vs blood at passage 4 and passage 30) that would reduce this widely held concern.

Response: I appreciate your insightful comment. As you mentioned, genome instability of human iPSCs raises safety concerns including genetic mutations impacting the p53 pathway and tumorigenicity. To assess the genome stability of our iPSC lines, we examined their karyotypes and short tandem repeat (STR) in previous studies (CRL-iPSCs; Patient #1-iPSCs and Patient #2-iPSCs)^{1,2}. Furthermore, we examined genome stability of ISC^{3D-hIO} during serial passaging (P8, P27, and P54) *in vitro* through the Whole Genome Sequencing (WGS). Based on the results of copy number variation (CNV) analysis, the genome of ISC^{3D-hIO} was maintained stably without any structural abnormalities after being cultured for ~6 months (27th passage, Fig. S3a). The genome of ISC^{3D-hIO}, which has been cultured for over a year (54th passage), has been generally stable, but we have observed chromosomal amplification in a specific region in chromosome 15 (Fig. S3b). Through the WGS analysis, we were able to confirm that stable *in vitro* cultivation of ISC^{3D-hIO} is possible for approximately 6 months. However, to enhance genomic stability during long-term passaging, we'll going to develop improved cultivation methods through further research.

a**P8 versus P27****b****P8 versus P54**
Fig S3. Whole-genome profiling of copy number variation (CNV) using whole genome short-read sequencing data. a. CNV difference between p8 and p27 samples. b. CNV difference between p8 and p54 samples.

Revised text in Results

To assess the genome stability of ISC^{3D-h10}, we performed whole-genome profiling of copy number variation (CNV) using whole genome short-read sequencing data (Extended Data Fig. S3). The genomic stability of ISC^{3D-h10} was well preserved for at least 6 months (P27) without structural variation (Extended Data Fig. S3a). While overall genome of ISC^{3D-h10} remained highly stable, amplified region was found in chromosome 15 at P54 (Extended Data Fig. S3b).

Revised text in Figure legends

Supplementary Fig. 3. Whole-genome profiling of copy number variation (CNV) using whole genome short-read sequencing data. a. CNV difference between p8 and p27 samples.
b. CNV difference between p8 and p54 samples.

Revised Materials and Methods

Whole genome sequencing

For whole genome sequencing of ISC^{3D-h10} (passage 8, 27, and 54), 100 ng of genomic DNA was used to construct DNA library with TruSeq Nano DNA (Illumina, USA) following the manufacturer's instruction. Multiple libraries were sequenced on an Illumina NovaSeq 6000 using paired-end 150, 6G reads. Reads were aligned to the reference genome Trimmomatic was used to remove low quality reads to reduce bias. Map the reads to the reference genome (hg38 from UCSC) of choice Burrows-Wheeler Aligner (BWA)²³. Properly mapped reads were extracted from BAM files after duplicated reads were removed. ngCGH (version 0.4.4) was used to compare two matched BAM data with a window size of 10 kb for copy number estimate. Then, the copy number altered regions were defined by segmentation of the genome using DNACopy (version 1.74.1)²⁴.

References

1. Jung, K.B. et al. Interleukin-2 induces the in vitro maturation of human pluripotent stem cell-derived intestinal organoids. *Nat Commun* **9**, 3039 (2018).
2. Kwon, O. et al. The development of a functional human small intestinal epithelium model for drug absorption. *Sci Adv* **7** (2021).
3. Elmentaite, R. et al. Single-Cell Sequencing of Developing Human Gut Reveals Transcriptional Links to Childhood Crohn's Disease. *Dev Cell* **55**, 771-783 e775 (2020).
4. Elmentaite, R. et al. Cells of the human intestinal tract mapped across space and time. *Nature* **597**, 250–255 (2021).
5. Yu, Q. et al. Charting human development using a multi-endodermal organ atlas and organoid models. *Cell* **184**, 3281-3298 e3222 (2021).
6. Spence, J.R. et al. Vertebrate intestinal endoderm development. *Dev Dyn* **240**, 501-520 (2011)
7. Kilpinen, H. et al. Common genetic variation drives molecular heterogeneity in human iPSCs. *Nat Commun* **546**, 370 (2017).
8. Guhr, A. et al. Recent trends in research with human pluripotent stem cells: Impact of research and use of cell lines in experimental research and clinical trials. *Stem Cell Rep* **11**, 485 (2018).
9. Jung, K.B. et al. *In vitro* and *in vivo* imaging and tracking of intestinal organoids from human induced pluripotent stem cells. *FASEB J.* **32**, 111 (2018).
10. Sugimoto, S. et al. Reconstruction of the human colon epithelium *in vivo*. *Cell Stem Cell* **22**, 171 (2018).
11. Zipeto, D. et al. ACE2/ADAM17/TMPRSS2 interplay may be the main risk factor for COVID-19. *Front Immunol* **11**, 576745 (2018).
12. Greco, V. et al. Histochemistry of the colonic epithelial mucins in normal subjects and in patients with ulcerative colitis. A qualitative and histophotometric investigation. *Gut* **8**, 491 (1967).
13. Martin, B. F. The goblet cell pattern in the large intestine. *Ant Rec* **140**, 1 (1961).
14. Delvin, E. E. et al. Developmental expression of calcitriol receptor, 9-kilodalton calcium-

- binding protein, and calcidiol 24-hydroxylase in human intestine. *Pediatr Res* **40**, 664 (1996).
15. Dunn, N. R. et al. Combinatorial activities of Smad2 and Smad3 regulate mesoderm formation and patterning in the mouse embryo. *Development* **131**, 1717 (2004).
 16. D'Amour, K. A. et al. Efficient differentiation of human embryonic stem cells to definitive endoderm. *Nat Biotech* **23**, 1534 (2005).
 17. Berardo, A. S. et al. BRACHYURY and CDX2 mediate BMP-induced differentiation of human and mouse pluripotent stem cells into embryonic and extraembryonic lineages. *Cell Stem Cell* **9**, 144 (2011).
 18. Spence, J. R. et al. Directed differentiation of human pluripotent stem cells into intestinal tissue in vitro. *Nature* **470**, 105 (2011)
 19. Tu, C. et al. Strategies for improving the maturity of human induced pluripotent stem cell-derived cardiomyocytes. *Cir Res* **123**, 512 (2018)
 20. Dedhia, P. H. et al. Organoid models of human gastrointestinal development and disease. *Gastroenterology* **150**, 1098 (2016)
 21. Munera, J. O. et al. Differentiation of human pluripotent stem cells into colonic organoids via transient activation of BMP signaling. *Cell Stem Cell* **21**, 51 (2017)
 22. Kim, J. et al. Human organoids: model system for human biology and medicine. *Nat Rev Mol Cell Biol* **21**, 571 (2020)
 23. Li, H. et al. Fast and accurate short read alignment with Burrows-Wheeler transformation. *Bioinformatics* **25**, 1754 (2009)
 24. Venkatraman, E. S. and Olsen, A. B. A Faster circular binary segmentation algorithm for the analysis of array CGH data. *Bioinformatics* **23**, 657 (2007)

REVIEWERS' COMMENTS

Reviewer #1 (Remarks to the Author):

The authors have done a thorough job with the revised manuscript and answered all my questions.

Reviewer #2 (Remarks to the Author):

The authors addressed the points that I raised in the previous review.

Reviewer #3 (Remarks to the Author):

My comments are addressed.

Regarding previous points raised by reviewer #4 and authors' response/revision:

Q1- has been answered in responses but cannot be in MS and specially queries like -"There also needs to be at least a discussion of why intestinal stem cells can be derived from iPSC/ESCs when this has not been possible for hematopoietic stem cells or epidermal stem cells (vast literature)." Cannot be part of a focus Study, such queries could be apt in reviews or general introduction.

Q2- has not been answered. authors just need to mention the exact catalogue No. and nomenclature of matrigel they use

Q3- I did not understand the query. As Normally we would replace a particular cell type with the same cell type, that is the purpose of Stem cell technology; so did not understand what the query meant -

'Is there a bias from this approach to yield ISCs that have properties of the small intestine? If so, how might this impact attempts at regenerative medicine where functionally distinct portions of the GI tract might be replaced with cells having small intestine properties?'

The authors did not plan to create a general GI stem cell.
Though authors have tried to reason it out, does not need to be addressed in MS

Q4. has been addressed suitable in MS and Responses.

RESPONSE TO REVIEWERS' COMMENTS

Reviewer #3 (Remarks to the Author):

My comments are addressed.

Regarding previous points raised by reviewer #4 and authors' response/revision:

Q1- has been answered in responses but cannot be in MS and specially queries like -"There also needs to be at least a discussion of why intestinal stem cells can be derived from iPSC/ESCs when this has not been possible for hematopoietic stem cells or epidermal stem cells (vast literature)." Cannot be part of a focus Study, such queries could be apt in reviews or general introduction.

Response: We appreciate your valuable comment. As you commented, the queries that are out of our focus will not be addressed in the manuscript.

Q2- has not been answered. authors just need to mention the exact catalogue No. and nomenclature of matrigel they use

Response: We appreciate your valuable comment. In all experiments, we used standardized formula Matrigel® Basement Membrane Matrix, LDEV-free, produced from Corning (Cat. No. 354234). The specific information of Matrigel was added in the Materials and Methods section.

Revised Materials and Methods

Differentiation of hPSC into hIO

The hIO differentiation method has previously been described in detail^{22, 23}. hPSCs were differentiated into definitive endoderm (DE) by treatment with 100 ng/ml activin A (R&D

Systems, Minneapolis, MN, USA) for three days in RPMI 1640 medium with increasing concentrations of 0%, 0.2%, and 2% defined foetal bovine serum (FBS; HyClone, **Cat. No. 16000044**, Thermo Fisher Scientific Inc., Waltham, MA, USA). DE cells were then treated for four days with RPMI 1640 medium containing 2% dFBS, 500 ng/ml FGF4 (**Peprotech, Cat. No.100-31-500**, Thermo Fisher Scientific Inc., Waltham, MA, USA), and 500 ng/ml WNT3A (**R&D Systems, Cat. No.5036-WN-500**, R&D Systems, Minneapolis, MN, USA) to promote differentiation into 3D hindgut spheroids. The spheroids were harvested and embedded in **Matrigel (Matrigel® Basement Membrane Matrix, LDEV-free, Cat. No. 354234, Corning, NY, USA)**, cultured in hIO medium composed of advanced Dulbecco's Modified Eagle Medium (DMEM)/F-12 medium (**Gibco, Cat No. 11330-099**, Thermo Fisher Scientific Inc., Waltham, MA, USA) containing 1× B27 (Invitrogen, **Cat No. 12587-010**, Thermo Fisher Scientific Inc., Waltham, MA, USA), 500 ng/ml R-Spondin 1 (**Peprotech, Cat. No. 120-38**), 100 ng/ml EGF (**R&D Systems, Cat. No. 236-EG-01M**), and 100 ng/ml Noggin (**R&D Systems, Cat. No. 6057-NG-01M**), and then passaged every two weeks. For *in vitro* maturation, hIO medium containing recombinant human interleukin 2 (**R&D Systems, Cat. No. 202-IL-010**) was used for two passages.

Q3- I did not understand the query. As normally we would replace a particular cell type with the same cell type, that is the purpose of Stem cell technology; so did not understand what the query meant -'Is there a bias from this approach to yield ISCs that have properties of the small intestine? If so, how might this impact attempts at regenerative medicine where functionally distinct portions of the GI tract might be replaced with cells having small intestine properties?'" The authors did not plan to create a general GI stem cell.

Though authors have tried to reason it out, does not need to be addressed in MS

Response: We appreciate your thoughtful consideration. As your suggestion, we will not address Q3 further in the manuscript.

Q4. has been addressed suitable in MS and Responses.